# The ubiquitin-conjugating enzyme UBE2D maintains a youthful proteome and ensures protein quality control during aging by sustaining proteasome activity

**Liam C. Hunt[1¤], Michelle Curley[1], Kudzai Nyamkondiwa[1], Anna Stephan[1], Jianqin Jiao[1], Kanisha Kavdia[2], Vishwajeeth R. Pagala[2], Junmin Peng[1,3], Fabio Demontis[1]***

**1** Department of Developmental Neurobiology, St. Jude Children's Research Hospital, Memphis, Tennessee, United States of America, **2** Center for Proteomics and Metabolomics, St. Jude Children's Research Hospital, Memphis, Tennessee, United States of America, **3** Department of Structural Biology, St. Jude Children's Research Hospital, Memphis, Tennessee, United States of America

¤ Current address: Department of Biology, Rhodes College, Memphis, Tennessee, United States of America
* Fabio.Demontis@stjude.org

**Data Availability Statement:** All data are available within the main text, Figs 1–7 and S1–S2, S1–S5 Data files, Source data S6 Data file and the S1 Raw

## Abstract

Ubiquitin-conjugating enzymes (E2s) are key for protein turnover and quality control via ubiquitination. Some E2s also physically interact with the proteasome, but it remains undetermined which E2s maintain proteostasis during aging. Here, we find that E2s have diverse roles in handling a model aggregation-prone protein (huntingtin-polyQ) in the *Drosophila* retina: while some E2s mediate aggregate assembly, UBE2D/effete (eff) and other E2s are required for huntingtin-polyQ degradation. UBE2D/eff is key for proteostasis also in skeletal muscle: eff protein levels decline with aging, and muscle-specific eff knockdown causes an accelerated buildup in insoluble poly-ubiquitinated proteins (which progressively accumulate with aging) and shortens lifespan. Mechanistically, UBE2D/eff is necessary to maintain optimal proteasome function: UBE2D/eff knockdown reduces the proteolytic activity of the proteasome, and this is rescued by transgenic expression of human UBE2D2, an eff homolog. Likewise, human UBE2D2 partially rescues the lifespan and proteostasis deficits caused by muscle-specific eff[RNAi] and re-establishes the physiological levels of eff[RNAi]-regulated proteins. Interestingly, UBE2D/eff knockdown in young age reproduces part of the proteomic changes that normally occur in old muscles, suggesting that the decrease in UBE2D/eff protein levels that occurs with aging contributes to reshaping the composition of the muscle proteome. However, some of the proteins that are concertedly up-regulated by aging and eff[RNAi] are proteostasis regulators (e.g., chaperones and Pomp) that are transcriptionally induced presumably as part of an adaptive stress response to the loss of proteostasis. Altogether, these findings indicate that UBE2D/eff is a key E2 ubiquitin-conjugating enzyme that ensures protein quality control and helps maintain a youthful proteome composition during aging.

Images. The TMT mass spectrometry proteomics data were deposited at the ProteomeXchange Consortium via the PRIDE partner repository and are accessible with the dataset identifiers PXD042345 and PXD045713.

**Funding:** Work in the Demontis lab is supported by the National Institute on Aging of the NIH (R01AG075869 to F.D.) and the Alzheimer's Association (AARG-NTF-22-973220 to F.D.). The Peng lab is supported by the NIH (RF1AG068581 to J.P.). The content is solely the responsibility of the authors and does not necessarily represent the official views of the National Institutes of Health. Research at St. Jude Children's Research Hospital is supported by the ALSAC (to F.D. and J.P.). The funders had no role in study design, data collection and analysis, decision to publish, or preparation of the manuscript.

**Competing interests:** The authors have declared that no competing interests exist.

**Abbreviations:** DIOPT, DRSC integrative orthology prediction tool; E1, ubiquitin-activating enzyme; E2, ubiquitin-conjugating enzyme; E3, ubiquitin ligase; GFP, green fluorescent protein; HD, Huntington's disease; HMW, high molecular weight; IP-MS, immunoprecipitation-mass spectrometry; RNAi, RNA interference; TFA, trifluoroacetic acid; TMT, tandem mass tag; UPS, ubiquitin/proteasome system.

## Introduction

Protein degradation regulates many cellular functions, and its derangement is the underlying cause of many human diseases [1–4]. Most proteins are regulated via degradation to ensure dynamic adjustments in their concentrations in response to cellular challenges and to maintain physiologic homeostasis. Moreover, damaged and misfolded proteins are degraded to avoid toxicity caused by their interaction with native proteins [1–4]. Because of these important functions, protein degradation is tightly controlled. In most cell types, the ubiquitin/proteasome system (UPS) is responsible for most of the protein degradation in the nucleus and cytoplasm, whereas the autophagy/lysosome system degrades cellular organelles, protein aggregates, and long-lived proteins [1–6]. In addition, several proteases and peptidases cooperate with the proteasome and autophagy in degrading target proteins [7–10].

To route proteins for degradation, the UPS relies on ubiquitination, a posttranslational modification that also regulates protein function and localization [11,12]. Through the concerted actions of a single ubiquitin-activating enzyme (E1), ~35 ubiquitin-conjugating enzymes (E2s), and ~620 E3 ubiquitin ligases, the UPS orchestrates the specific poly-ubiquitination of protein substrates in humans, which can lead to their degradation [6,13–16]. Depending on the ubiquitin lysine residue employed to build poly-ubiquitin chains, poly-ubiquitinated proteins can be preferentially degraded by the proteasome or by the autophagy-lysosome system [11,12,17–21]. E2s have a pivotal role in the ubiquitination cascade and direct the recruitment of E3 ubiquitin ligases and target proteins [21–24].

In addition to regulating normal protein turnover, E2s can also ensure proteostasis. For example, the E2 enzyme UBE2B and its associated E3 ligase UBR4 were found to regulate muscle protein quality during aging in *Drosophila* and mice [25,26]. The capacity of E2s to maintain proteostasis may derive from their role in guiding the ubiquitination and proteasome-mediated degradation of misfolded and aggregation-prone proteins [27,28] and to trigger autophagy [29,30]. Moreover, because some E2s are physically associated with the proteasome, it was proposed that they may directly regulate its proteolytic activity [31–33]. For example, the E2 enzymes Ubc1/2/4/5 interact with the proteasome and this association further increases with heat stress in yeast [31], a condition where proteasome function is modulated by ubiquitination of proteasome components, such as Rpn13, mediated by the proteasome-associated E3 UBE3C [34]. While E2s may regulate proteostasis via several mechanisms, it remains largely undetermined which E2s are crucial for maintaining protein quality control during aging.

Here, we have utilized a model aggregation-prone protein (GFP-tagged pathogenic huntingtin, Htt-polyQ-GFP) expressed in the *Drosophila* retina to test the impact of E2 ubiquitin-conjugating enzymes on proteostasis. In addition, we have similarly tested a set of E3 ubiquitin ligases and deubiquitinating enzymes (DUBs) that were identified by mapping the E2 interactome in human cells [35]. These analyses reveal diverse functions of E2s and associated enzymes in proteostasis. Contrary to the expectation that E2s may generally promote the proteasomal degradation of Htt-polyQ and hence reduce Htt-polyQ levels, we find that the knockdown of several E2s has the opposite effect, i.e., it decreases Htt-polyQ aggregates.

Knockdown of the E2 enzyme effete (eff), homologous to human UBE2D1/2/3/4 ubiquitin-conjugating enzymes, is one of the relatively few RNAi interventions that increase the amount of Htt-polyQ aggregates, and we find that this is due to an impediment in Htt-polyQ protein degradation. Consistent with a prevalent role of UBE2D/eff in proteostasis across aging tissues in *Drosophila*, we find that UBE2D/eff protein levels significantly decline during aging in skeletal muscle, and that muscle-specific eff knockdown increases the age-associated accumulation of insoluble poly-ubiquitinated proteins and reduces lifespan. Interestingly, the proteostasis deficits caused by UBE2D/eff knockdown are partially rescued by transgenic expression of one

of its human homologs, UBE2D2, suggesting that UBE2D ubiquitin-conjugating enzymes have evolutionary conserved roles in maintaining protein quality control during aging. Altogether, our study identifies diverse roles for E2 ubiquitin-conjugating enzymes in handling aggregation-prone proteins and managing proteostasis and that UBE2D/eff is key for ensuring protein quality control during aging.

## Results

### Diverse roles of E2 ubiquitin-conjugating enzymes in the disposal of pathogenic huntingtin

Huntington's disease (HD) is caused by pathogenic huntingtin with polyQ tract expansion (Htt-polyQ), a toxic, aggregation-prone protein that induces neurodegeneration [36–38]. Previous studies have shown that ubiquitination is a conserved regulator of polyglutamine aggregation. Loss of E2s has been shown to have a variable impact on the size and number of Htt-polyQ aggregates: aggregate density is increased by the loss of some E2s but is decreased by the loss of others [25,26,39,40]. However, as for other analyses of E2 function, a complete understanding of the role of E2s in polyglutamine aggregation is missing.

The fruit fly *Drosophila melanogaster* is an ideal model for studying the mechanisms of Htt-polyQ protein degradation. GFP-tagged Htt-polyQ72 can be expressed in the fly retina [41] and fluorescent Htt-polyQ-GFP aggregates can be visualized from intact animals over time [42,43]. In particular, the fly retina is a convenient tissue to examine Htt-polyQ because this tissue is dispensable for fly survival. With this model, it was previously found that there is an age-dependent increase in the amount of Htt-polyQ72 protein aggregates, which reflects the overall age dependency of HD phenotypes also observed in humans [36,37].

To test whether E2s regulate Htt-polyQ-GFP aggregation during aging, multiple RNAi lines were utilized to target each of the 21 *Drosophila* E2s. For these studies, transgenic RNAi was driven in the *Drosophila* retina with the UAS/Gal4 system and GMR-Gal4, concomitantly with Htt-polyQ-GFP (Fig 1A). After 30 days at 29˚C, the total area of Htt-Q72-GFP fluorescent aggregates was scored with CellProfiler and compared to control RNAi interventions (Fig 1A–1C and S1 Data). Overall, there was a trend towards decreased aggregate area upon knockdown of several E2s, including the *Drosophila* homologs of UBE2F/M, UBE2I, UBE2QL1, and UBE2Z, suggesting that these E2s may be required for ubiquitination-dependent aggregation of Htt-polyQ-GFP (Fig 1A and 1B). Analyses of deubiquitinating enzymes (DUBs) and E3 ubiquitin ligases identified from mapping the E2 interactome in human cells [35] indicate that RNAi for these interactors yields overall similar phenotypes as those found by knocking down the interacting E2s (Fig 1A–1C). For example, RNAi for E3s and DUBs that interact with Ubc10 (UBE2L3) and CG7656 (UBE2R1/2) reduce the area of Htt-polyQ-GFP aggregates, as found for the knockdown of these E2s (Fig 1C). The decreased polyglutamine aggregation that is observed upon E2 RNAi may indicate that E2-mediated ubiquitination is required for Htt aggregate formation. Consistent with this model, it was previously reported that K48 linkage-specific ubiquitination drives Htt degradation by the proteasome, whereas K63 ubiquitination promotes Htt aggregation [44]. However, the decline in Htt-polyQ-GFP aggregates may also result from reduced transgenic expression of Htt-polyQ-GFP. This appears to occur with the knockdown of UBE2K/Ubc4 and UBE2Z/Bruce, whereas there is a partial increase in Htt-polyQ-GFP mRNA levels with UBE2QL1/CG4502 RNAi (S1 Fig). Mechanistically, these E2s may modulate transgenic expression by regulating histone ubiquitination and degradation (which generally alters transcriptional activity), as previously shown for the Ubc4 ortholog UBE2K [45]. Altogether, our comprehensive analysis highlights the differential effects of

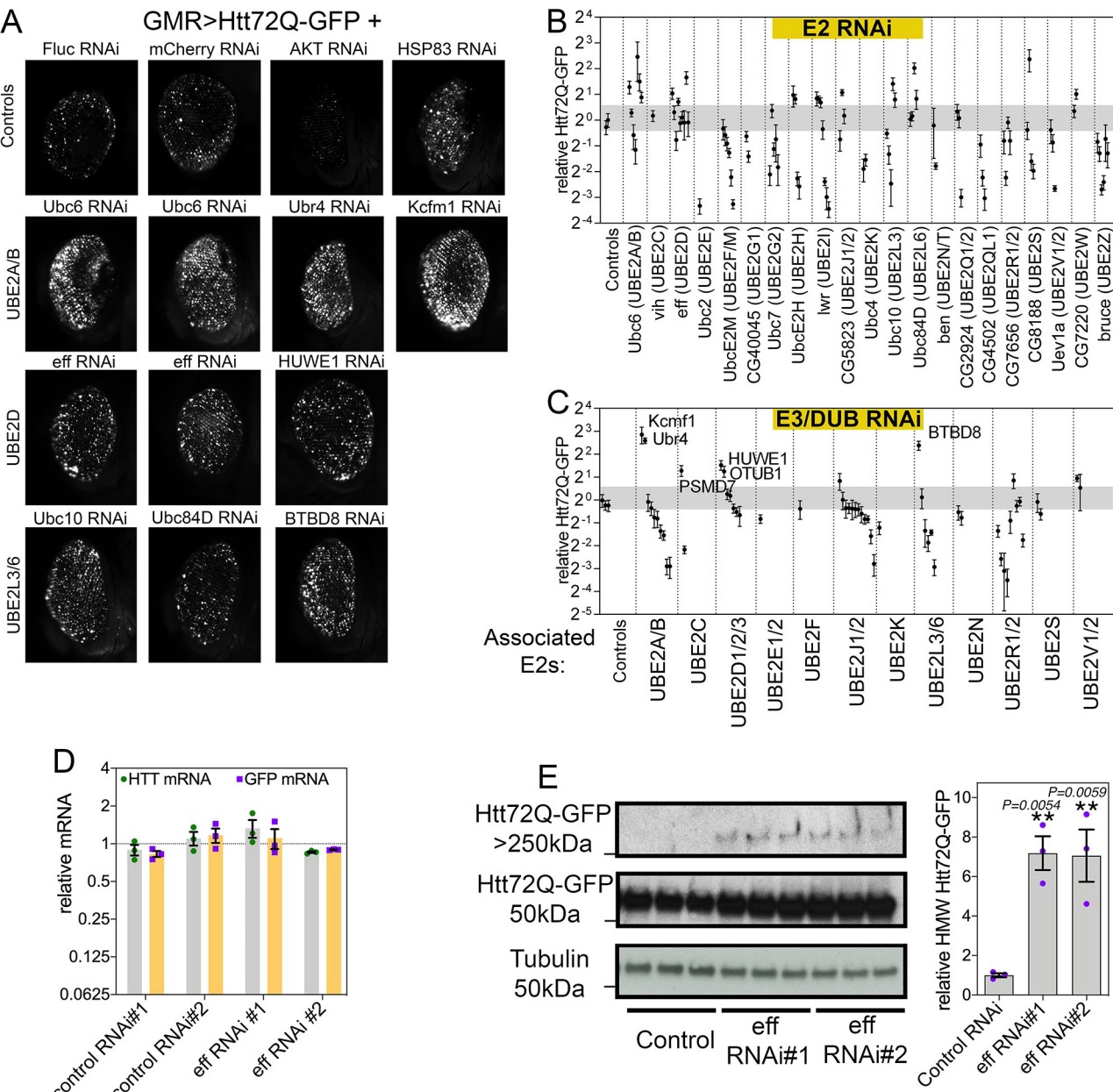

**Fig 1. RNAi for E2 ubiquitin-conjugating enzymes and associated E3s and DUBs modulates pathogenic huntingtin-polyQ aggregates in the _Drosophila_ retina.** (**A**) GFP-tagged huntingtin-polyQ (Htt-polyQ72-GFP) driven with _GMR-Gal4_ leads to GFP-fluorescent Htt protein aggregates in the retina at 30 days of age. Compared to negative controls (mcherry[RNAi] and luciferase[RNAi]), RNAi for E2 ubiquitin-conjugating enzymes, associated deubiquitinating enzymes (DUBs), and E3 ubiquitin ligases (E3s) modulates the amount of Htt-polyQ72-GFP aggregates. Specifically, RNAi for Ubc6 (homologous of UBE2A/B) and for its associated E3 ubiquitin ligases Ubr4 and Kcmf1 increases the amount of Htt-polyQ72-GFP protein aggregates. Similar increases are also seen with the knockdown of UBE2D/eff and the associated E3 HUWE1, and with RNAi for Ubc10/UBE2L3, Ubc84D/ UBE2L6, and for the associated E3 enzyme BTBD8. Positive controls include Akt RNAi, which reduces protein aggregates, and Hsp83 RNAi, which increases them. (**B**, **C**) Quantitation of the total area of Htt-polyQ72-GFP aggregates modulated by RNAi for E2s (B) and for associated E3s and DUBs (C). Relative fold changes compared to control RNAi interventions are shown; $n = 5$ (biological replicates), SD. Each data point in the graph represents a single RNAi targeting the corresponding gene. For each RNAi, the mean of 5 biological replicates is shown, with each biological replicate representing a single eye (each from a distinct animal). The variability in the analysis of each single gene is likely due to the distinct efficacy of the different RNAi lines targeting the same gene. (**D**) qRT-PCR indicates that there are no changes in _GFP_ and _Htt_ mRNA levels upon eff knockdown compared to control RNAi, indicating that eff RNAi does not modulate the amount of GFP-tagged huntingtin-polyQ aggregates via changes in the expression of _Htt-polyQ72-GFP_ transgenes; $n = 3$ (biological replicates) and SEM. (**E**) Levels of Htt-polyQ72-GFP aggregates detected by western blot with anti-GFP antibodies identify Htt-polyQ72-GFP monomers (~50 kDa) and HMW assemblies of Htt-polyQ72-GFP in the stacking gel (>250 kDa). Knockdown of UBE2D/eff increases the levels of HMW Htt-polyQ72-GFP; $n = 3$ (biological replicates), SEM, and $p$-values (one-way ANOVA) are indicated, with **$p < 0.01$, compared to mcherry[RNAi]. See also S1 Data. The data underlying the graphs shown in this figure can be found in the S6 Data file. Uncropped western blots are available in S1 Raw Images.

distinct E2s on Htt disposal and the similarity in the response induced by the E3s and DUBs interacting with each E2.

## UBE2D/eff knockdown increases aggregates of pathogenic huntingtin in the *Drosophila* retina

In addition to RNAi lines that decrease the amount of Htt aggregates, we found that RNAi-mediated knockdown of other E2s causes an increase in Htt aggregates (Fig 1A–1C). Specifically, RNAi for Ubc6 (UBE2A/B), eff (UBE2D), Ubc10 (UBE2L3), and Ubc84D (UBE2L6) increased the total area of protein aggregates. Likewise, RNAi for E3/DUBs that physically associate with these E2s (such as Ubr4 and Kcfm1, associated with UBE2A/B; and HUWE1, associated with UBE2D) led to higher levels of polyglutamine aggregates. A possible interpretation of these results is that these E2s are necessary for the degradation of aggregation-prone Htt and, consequently, their loss leads to the accumulation of detergent-insoluble polyglutamine aggregates. To test this hypothesis, we further characterized the molecular mechanisms by which UBE2D/eff regulates Htt proteostasis. We first measured by qRT-PCR the levels of Htt-polyQ-GFP and found that there are no changes in its expression upon eff RNAi compared to controls (Fig 1D), indicating that increased levels of Htt-polyQ-GFP aggregates observed upon eff RNAi do not arise from increased transgenic expression of Htt-polyQ-GFP. Next, we utilized western blots to analyze the levels of Htt-polyQ-GFP monomers (~50 kDa) and high molecular weight (HMW) Htt-polyQ-GFP aggregates that accumulate in the stacking gel (>250 kDa). Remarkably, 2 different RNAi lines targeting UBE2D/eff increase HMW Htt-polyQ-GFP levels (Fig 1E), suggesting that UBE2D/eff is required for Htt degradation. In summary, these findings indicate an extensive role of E2 ubiquitin-conjugating enzymes in modulating Htt-Q72-GFP aggregation during aging in the *Drosophila* retina.

## Retinal degeneration caused by UBE2D/eff knockdown is rescued by human UBE2D2/4

Compared to controls, we find that RNAi for Ubc6 and for eff induces retinal degeneration, as indicated by the "rough eye" appearance (indicative of photoreceptor death and derangement of inter-ommatidial bristles [46,47]), the increase in the eye area occupied by necrotic patches, and/or the loss of pigmentation (Fig 2A). On this basis, we next examined whether concomitant transgenic expression of the human homologs of these *Drosophila* E2s rescues retinal degeneration. To this purpose, we expressed human UBE2B (homologous to Ubc6, DIOPT homology score = 13) and human UBE2D2 and UBE2D4 (homologous to eff, DIOPT scores of 13 and 10, respectively) and compared their capacity to rescue retinal degeneration compared to mock mcherry overexpression (Fig 2A). As expected based on the sequence homology, hUBE2B (but not mcherry, hUBE2D2, and hUBE2D4) rescued retinal degeneration (i.e., necrotic patches and depigmentation) induced by Ubc6 RNAi. Similarly, hUBE2D2 and to a lower extent hUBE2D4 rescued depigmentation caused by eff knockdown. Altogether, these findings indicate that Ubc6 and eff loss can be rescued by transgenic expression of their respective human homologs UBE2B and UBE2D2/4 (Fig 2A).

On this basis, we next examined whether hUBE2D can rescue the defects in Htt-polyQ proteostasis caused by eff RNAi. Specifically, we compared eff RNAi versus control RNAi, with the concomitant expression of either mcherry or hUBE2D2. Image analysis of Htt-poly-GFP aggregates indicates that hUBE2D2 impedes the accumulation of Htt-polyQ-GFP aggregates which otherwise increase with eff RNAi (Fig 2B). Similar results were also obtained by western blot: there was a lower accumulation of HMW Htt-polyQ-GFP in eff$^{RNAi}$+hUBE2D2 versus eff$^{RNAi}$+mcherry and the other controls (Fig 2C). In summary, these findings indicate that

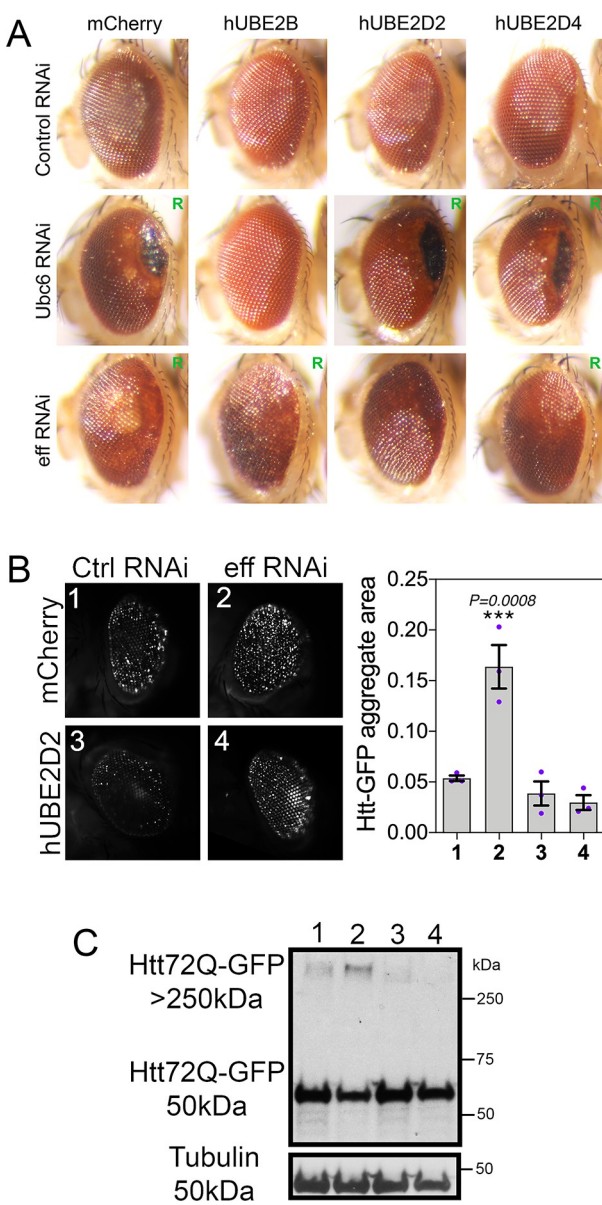

**Fig 2. Retinal degeneration induced by knockdown of Ubc6 and eff is rescued by expression of the respective human homologs UBE2B and UBE2D2/4.** (**A**) Compared to control RNAi, knockdown of Ubc6 and eff causes retinal degeneration, as indicated by the "rough eye" phenotype (R), by black necrotic patches, and/or by areas with depigmentation. Human UBE2B rescues degeneration induced by RNAi for its *Drosophila* homolog Ubc6 but not degeneration induced by RNAi for the unrelated E2 enzyme eff. Likewise, human UBE2D2 and UBE2D4 rescue retinal degeneration induced by knockdown of their *Drosophila* homolog eff. (**B**) Htt-polyQ72-GFP aggregates increase in *Drosophila* retinas with eff RNAi and this is rescued by human UBE2D2 but not by mcherry; $n = 3$ (biological replicates), SEM, and *p*-values (one-way ANOVA) are indicated, with ***$p < 0.001$. (**C**) Similarly, HMW assemblies of Htt-polyQ72-GFP detected in the stacking gel increase with eff RNAi compared to control RNAi but this is rescued by hUBE2D2 versus control mcherry expression. The data underlying the graphs shown in this figure can be found in the S6 Data file. Uncropped western blots are available in S1 Raw Images.

UBE2D/eff has an evolutionary-conserved function in maintaining proteostasis in an HD model.

## UBE2D/eff protein levels decline with aging, and UBE2D/eff knockdown impairs skeletal muscle proteostasis as observed during aging

The comprehensive analysis reported above has revealed a range of diverse roles for E2s in modulating polyglutamine aggregates in *Drosophila*. UBE2D/eff appears to be one of the E2s with the most striking effects in preserving proteostasis, i.e., it is required for preventing the accumulation of HMW Htt-polyQ-GFP (Figs 1 and 2). On this basis, we further probed the role of UBE2D/eff in protein quality control. To this purpose, we examined skeletal muscle aging, which is characterized by a progressive decline in proteostasis and by the age-related accumulation of detergent-insoluble poly-ubiquitin protein aggregates [25,48–54].

Deep-coverage tandem mass tag (TMT) mass spectrometry [55,56] of skeletal muscle from young and old control w[1118] flies (1 and 8 weeks old) detected 5,971 proteins, including 10 of the 21 *Drosophila* E2 ubiquitin-conjugating enzymes (Fig 3A and S2 Data). Interestingly, UBE2D/eff displayed the strongest age-dependent down-regulation (Fig 3B), suggesting that a reduction in UBE2D/eff protein levels may contribute to the progressive loss of protein quality control with aging.

To test this hypothesis, we experimentally reduced UBE2D/eff levels from a young age by driving UBE2D/eff RNAi with MhcF3-Gal4 [25] in skeletal muscle and compared this intervention to a control mcherry RNAi. Immunostaining and confocal microscopy of skeletal muscle of flies at 3 weeks of age indicate that UBE2D/eff RNAi significantly increases the size of poly-ubiquitin protein aggregates and that there is an overall trend towards a higher total area of aggregates, compared to control RNAi (Fig 3C and 3D). Altogether, these results indicate that anticipating the age-related decline in UBE2D/eff protein levels deranges proteostasis.

We next examined muscle proteostasis via the analysis of detergent-soluble and insoluble fractions [57]. Knockdown of UBE2D/eff with 2 distinct RNAi lines increased the levels of poly-ubiquitinated proteins found in both the soluble and insoluble fractions of skeletal muscle of 3-week-old flies. There was also an increase in p62/Ref(2)P levels that occurred independently of mRNA changes (S2 Fig) and in parallel with the accumulation of poly-ubiquitinated proteins, although this occurred primarily in the detergent-soluble fraction (Fig 3E and 3F). Altogether, these findings indicate that UBE2D/eff is required to ensure protein quality control in skeletal muscle.

## Defects in proteostasis and lifespan caused by muscle-specific UBE2D/eff knockdown are partially rescued by transgenic expression of human UBE2D2

We have found that human UBE2D2 can rescue retinal degeneration induced by eff RNAi in the context of polyglutamine disease, and that this results from the degradation of HMW Htt-polyQ (Fig 2). On this basis, we next examined whether hUBE2D2 can also rescue defects in muscle proteostasis due to eff knockdown. To this purpose, we examined detergent-soluble and insoluble fractions from muscles of flies with eff[RNAi]+hUBE2D2 versus eff[RNAi]+mcherry as well as control[RNAi]+hUBE2D2 versus control[RNAi]+mcherry (Fig 4A and 4B). Western blot analyses with anti-ubiquitin and anti-p62/Ref(2)P antibodies revealed that eff[RNAi]+hUBE2D2 reduced the detergent-soluble levels of ubiquitinated proteins and p62/Ref(2)P compared to eff[RNAi]+mcherry and other controls, whereas there was no rescue of detergent-insoluble levels (Fig 4A and 4B).

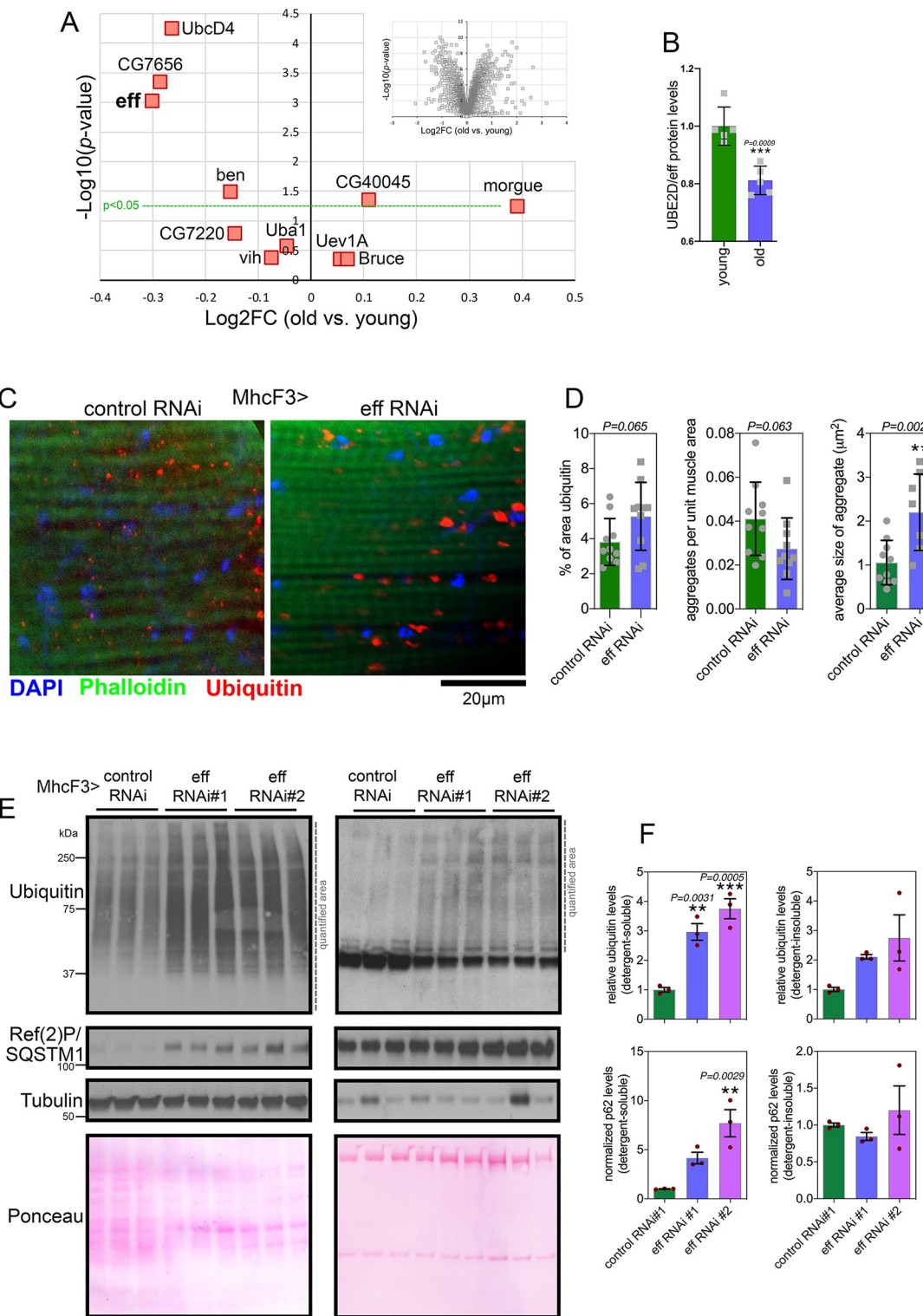

**Fig 3. RNAi for the E2 enzyme UBE2D/eff impairs muscle protein quality control during aging. (A)** TMT mass spectrometry of *Drosophila* skeletal muscle from old versus young w[1118] flies at 8 vs. 1 week-old; $n$ = 5 biological replicates. The $x$-axis reports the $\log_2$FC, whereas the $y$-axis reports the $-\log_{10}(p$-value). The E1 enzyme UBA1 and the E2 enzymes that are detected are indicated, including UBE2D/eff; $n$ = 5 (biological replicates). See also S2 Data. **(B)** The protein levels of UBE2D/eff significantly decline with aging (8 vs. 1 week old) in skeletal muscle; $n$ = 5 (biological replicates), SD, and $p$-values (Student's $t$

test). (**C, D**) Immunostaining of *Drosophila* skeletal muscle at 3 weeks of age indicates that UBE2D/eff RNAi impairs protein quality control, as indicated by the higher age-related accumulation of aggregates of poly-ubiquitinated proteins, compared to control RNAi. The scale bar is 20 μm. In (C), *n* = 10 (biological replicates), SD, and *p*-values (Student's *t* test). (**E, F**) Western blot analysis of detergent-soluble and insoluble fractions from *Drosophila* skeletal muscle indicates that eff RNAi impedes proteostasis, as indicated by higher levels of detergent-soluble and insoluble poly-ubiquitinated proteins compared to control RNAi (E). A similar increase is also found for the detergent-soluble levels of p62/Ref(2)P (E). *n* = 3 (biological replicates), SEM, **$p < 0.01$, ns = not significant (one-way ANOVA). The data underlying the graphs shown in this figure can be found in the S6 Data file. Uncropped western blots are available in S1 Raw Images.

Because muscle proteostasis can regulate organismal aging [25,48,50,58,59], we next determined the impact of muscle-specific UBE2D/eff loss on lifespan. Knockdown of eff with 2 distinct RNAi lines (Fig 4C) shortened lifespan (Fig 4D) and this was partially rescued by concomitant expression of hUBE2D but not by control mcherry (Fig 4E). Altogether, these studies identify a key role for UBE2D/eff in proteostasis during skeletal muscle aging in *Drosophila* and highlight the evolutionary conservation of UBE2D function.

## UBE2D/eff is necessary to maintain optimal proteasomal activity in skeletal muscle

In addition to participating in the ubiquitination cascade, there is a growing understanding that mediators of ubiquitination (such as E2 and E3 enzymes) can also directly associate with the proteasome and regulate proteasome activity [26,32,34,60]. In particular, UBE2D/eff is homologous to 2 related E2 ubiquitin-conjugating enzymes (*S. cerevisiae* UBC4 and UBC5; DIOPT scores = 14 and 13) that were previously reported to associate with the proteasome in yeast [31]. On this basis, we next monitored whether the proteolytic activity of the proteasome is modulated by UBE2D/eff RNAi and found it to be the case: UBE2D/eff RNAi significantly reduces the trypsin-like proteasome activity and also substantially reduces the chymotrypsin-like activity. Importantly, such eff[RNAi]-induced decline in proteasome activity is rescued by the coexpression of human UBE2D2 but not by a control transgene (Fig 5A–5C).

Such eff[RNAi]-induced decline in proteasome activity occurs without major changes in proteasome levels. Specifically, while TMT mass spectrometry detects an average approximately 15% increase in the levels of proteasome components in response to UBE2D/eff[RNAi] (S3 Data), western blot analyses with antibodies for components of the 19S regulatory particle (p42A/Rpn7) and for the 20S catalytic core (α subunits) of the proteasome indicate that there are no substantial changes (Fig 5D–5G). Because the proteasome can transition to insoluble aggregates in response to stress and aging [61,62], we also monitored its insoluble levels but found that they do not change in response to eff[RNAi] (Fig 5D–5G).

Although autophagy can be induced as a backup mechanism to cope with proteasome stress [63,64], eff[RNAi] does not modulate Atg8 mRNA levels (S2 Fig). Moreover, eff[RNAi] does not affect the total protein levels of LC3/Atg8 and its processing from Atg8-I to Atg8-II (Fig 5D and 5F) as assessed from the western blot analysis of detergent-soluble fractions, suggesting that autophagy is not substantially regulated. There was, however, a decline in the insoluble Atg8-II/Atg8-I levels upon eff[RNAi] (Fig 5E and 5G), which may indicate subtle changes in autophagy. Altogether, these findings indicate that UBE2D/eff maintains optimal proteasome activity without substantially perturbing autophagy.

## Deep-coverage proteomics reveals protein substrates that are modulated by UBE2D/eff

To determine the proteins that are modulated by UBE2D/eff RNAi and the partial rescue of proteostasis defects by hUBE2D, TMT-based proteomics [55,56] was utilized to determine the

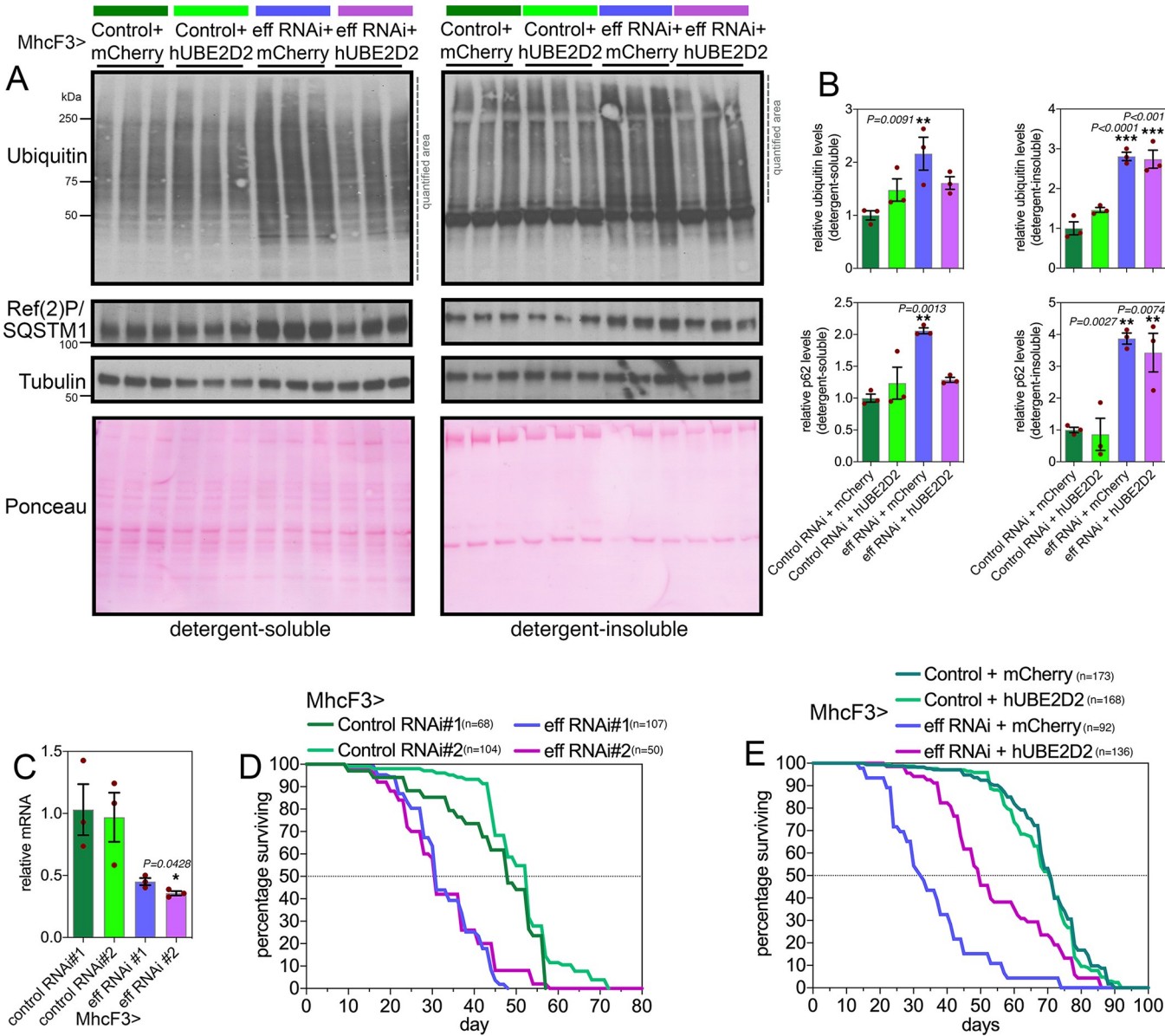

**Fig 4. Defects in proteostasis and lifespan caused by muscle-specific UBE2D/eff RNAi are partially rescued by the expression of human UBE2D2. (A, B)** Western blot analysis of detergent-soluble and insoluble fractions from *Drosophila* skeletal muscle indicates that defects in proteostasis due to eff RNAi can be partially rescued by its human homolog UBE2D2, as indicated by the normalization of the detergent-soluble levels of poly-ubiquitinated proteins and p62/Ref (2)P. However, hUBE2D2 does not impact their detergent-insoluble levels, indicating that hUBE2D only partially rescues defects induced by UBE2D/eff RNAi. In (B), *n* = 3 (biological replicates), SEM, **p < 0.01 (one-way ANOVA). **(C)** Muscle-specific UBE2D/eff RNAi reduces eff mRNA levels; *n* = 3 (biological replicates), SEM, **p < 0.01 (one-way ANOVA). **(D)** Two distinct RNAi lines targeting UBE2D/eff reduce lifespan compared to control RNAi (*p* < 0.001, log-rank test). **(E)** The decline in organismal survival due to UBE2D/eff RNAi in muscle is partially rescued by transgenic hUBE2D expression compared to control mcherry overexpression (*p* < 0.001, log-rank test). The data underlying the graphs shown in this figure can be found in the S6 Data file. Uncropped western blots are available in S1 Raw Images.

protein changes modulated by UBE2D/eff RNAi versus control RNAi, and whether these are rescued by hUBE2D2 compared to control mcherry expression [35]. Analysis of eff$^{RNAi}$-induced protein changes revealed several categories that are enriched among the significantly regulated proteins (Fig 6A–6D and S3 Data; [35]). Up-regulated proteins included proteasome components, chaperones, deubiquitinases, ubiquitin ligases, and extracellular proteins,

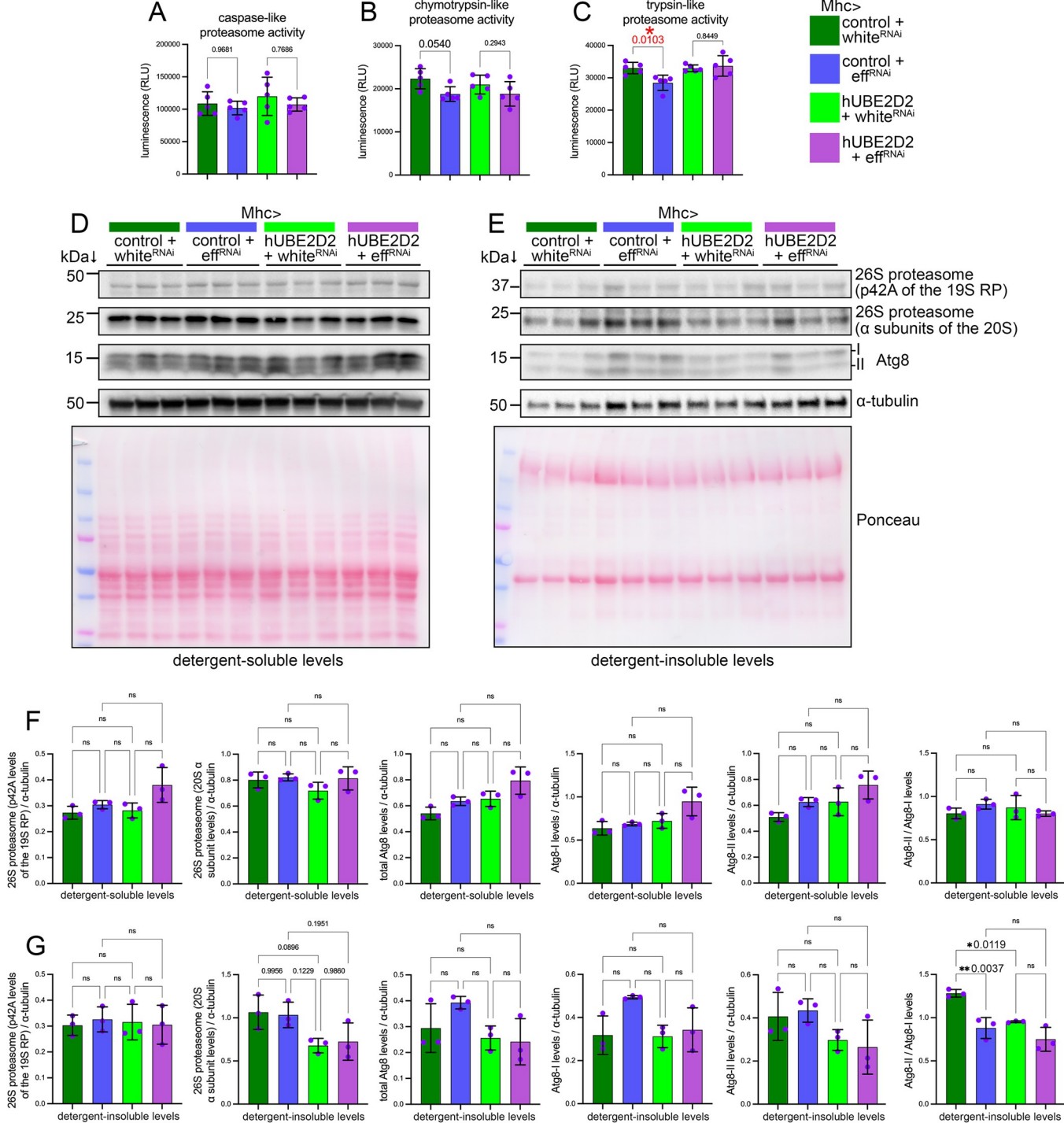

**Fig 5. Muscle-specific knockdown of UBE2D/eff reduces the proteolytic activity of the proteasome.** (**A–C**) Proteasome activity assays from the thoraces of 14-day-old flies with muscle-specific knockdown of UBE2D/eff or mock, and with or without rescue by human UBE2D2. Measurement of the proteolytic activities of the proteasome indicates that UBE2D/eff RNAi significantly reduces the trypsin-like proteasome activity ($p = 0.0103$), and there is also a trend for reducing the chymotrypsin-like activity ($p = 0.054$). Expression of human UBE2D2 rescues the reduction in proteasome activity compared to a control transgene. The graphs display the mean ± SD with $n = 5$ (biological replicates). Statistical analysis was done with one-way ANOVA with Sidak's multiple comparisons test; $^*p < 0.05$. (**D–G**) Western blot analysis of detergent-soluble (D) and insoluble (E) fractions from the thoraces of 14-day-old flies with muscle-specific knockdown of UBE2D/eff or mock and with or without rescue by human UBE2D2. There are no significant changes in the soluble (F) and insoluble (G) levels of the 26S proteasome, as ascertained with antibodies raised against the p42A/Rpn7 component of the 19S regulatory particle (RP) and against the α subunits of the 20S catalytic core of the *Drosophila* proteasome. However, TMT mass spectrometry detects an average 15% increase in the levels of proteasome

components (S3 Data). There are no significant changes in the total levels of Atg8, Atg8-I, Atg8-II, and Atg8-II/Atg8-I in the soluble fractions, indicating that UBE2D/eff does not substantially impact Atg8 levels and processing. There was, however, a significant decline in the insoluble Atg8-II/Atg8-I levels upon eff^RNAi, which may indicate subtle changes in autophagy. The graphs display the mean ± SD with $n = 5$ (biological replicates). Statistical analysis was done with one-way ANOVA with Tukey's multiple comparisons test; *$p < 0.05$, **$p < 0.01$. The data underlying the graphs shown in this figure can be found in the S6 Data file. Uncropped western blots are available in S1 Raw Images.

whereas down-regulated proteins were enriched for peptidases, lipases, and secreted proteins (Fig 6E).

Curated analyses revealed that, as expected, eff protein levels declined in response to eff RNAi compared to control RNAi (Fig 6F). The levels of several other proteins were increased upon eff RNAi but reduced by concomitant rescue with hUBE2D2 (Fig 6F). These proteins include Arc1 and Arc2 (activity-regulated cytoskeleton-associated proteins 1 and 2), which regulate starvation-induced locomotion, the neuromuscular junction, and metabolism [65–67]; the glycine N-methyltransferase Gnmt enzyme, which generates sarcosine and controls the amount of the methyl donor S-adenosylmethionine [68,69]; and CG4594, which encodes for an enzyme involved in fatty acid beta-oxidation. In addition to rescuing protein changes induced by eff RNAi, overexpression of hUBE2D2 by itself reduced the levels of Arc1, Arc2, Gnmt, and CG4594. Altogether, these proteomics analyses identify protein targets that are modulated by UBE2D/eff in skeletal muscle (Fig 6F). Improper UBE2D/eff-mediated turnover of these proteins and their consequent accumulation may contribute to deranging muscle protein quality control and in turn to reduced survival. In agreement with this hypothesis, it was previously found that elevation of Arc1 protein levels in Alzheimer's disease models causes cytotoxicity and contributes to neuronal death in *Drosophila* [70].

Conversely, other proteins were induced in muscles with UBE2D/eff knockdown and may represent a protective response that mitigates the derangement of proteostasis due to UBE2D/eff knockdown. These included Pomp, involved in proteasome assembly [71,72], the chaperones Hsp22, Hsp23, and Hsp26, and the ubiquitin-like protein UBL3 (Fig 6G). In agreement with this scenario, gene expression analysis indicates that eff RNAi up-regulates the mRNA levels of *Pomp*, *Hsp22*, *Hsp26*, and *Hsp70* (S2 Fig). These findings suggest that a stress response (presumably related to the unfolded protein response or the proteasome stress response [59,73–75]) is triggered by the decline in protein quality control due to UBE2D/eff knockdown. In summary, these studies indicate that UBE2D/eff knockdown induces several proteomics changes, some of which are rescued by hUBE2D2. While some of these changes may be cytotoxic (e.g., increased Arc1 levels, [70]), others may conversely limit the derangement of protein quality control and/or improve survival.

## RNAi for UBE2D/eff causes proteomic changes associated with aging in *Drosophila* skeletal muscle

We have found that UBE2D/eff protein levels decline during aging in skeletal muscle (Fig 3A and 3B) and that the experimental induction of UBE2D/eff knockdown at a young age induces a premature decline in proteostasis (Figs 3 and 4), which otherwise is normally lost only in old age [48,49]. Lastly, we have found that UBE2D/eff^RNAi modulates the levels of many proteins (Fig 6). On this basis, we next examined how the proteomic changes induced by aging correlate with those induced by eff RNAi in young age. To this purpose, we cross-compared the TMT mass spectrometry data sets obtained from *Drosophila* thoraces (which consist primarily of skeletal muscle) with muscle-specific eff RNAi (normalized by control mcherry RNAi) and those obtained from control thoraces of old versus young flies. Cross-comparison of the significantly regulated proteins revealed that 700 out of 1,002 proteins (approximately 70%) are

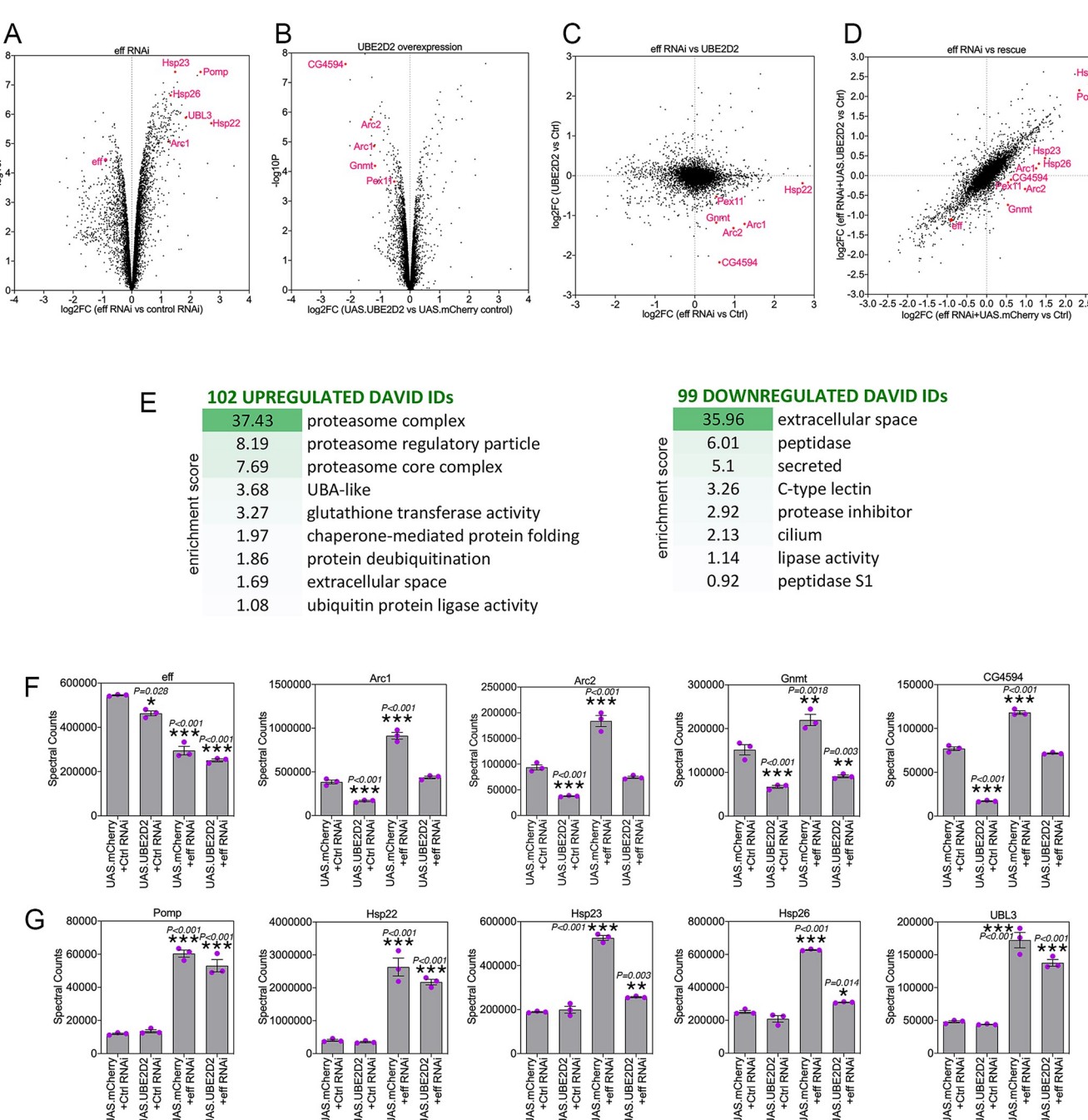

**Fig 6. Deep-coverage TMT mass spectrometry identifies the proteins modulated by UBE2D/eff knockdown in *Drosophila* skeletal muscle.** (**A, B**) TMT mass spectrometry of *Drosophila* skeletal muscle with UBE2D/eff knockdown vs. control RNAi (A) and overexpression of human UBE2D2 vs. control mcherry (B). The *x*-axis reports the log$_2$FC, whereas the *y*-axis reports the–log$_{10}$(*p*-value). Examples of regulated proteins are shown in red. (**C**) Cross-comparison of the log$_2$FC induced by eff RNAi vs. control RNAi (*x*-axis) with the log$_2$FC of hUBE2D2 vs. control mcherry (*y*-axis). (**D**) Cross-comparison of the log$_2$FC induced by eff RNAi + mcherry vs. control (*x*-axis) with the log$_2$FC of eff RNAi + hUBE2D2 vs. control (*y*-axis) indicates that some of the protein changes induced by eff RNAi are rescued by hUBE2D2. (**E**) GO term analysis of protein categories that are enriched among up-regulated and down-regulated proteins in response to UBE2D/eff RNAi. The enrichment scores and number of regulated DAVID IDs are shown. (**F, G**) Examples of regulated proteins modulated by eff RNAi and hUBE2D-mediated rescue include protein changes that may drive derangement of proteostasis, such as Arc1/2 up-regulation (F), as well as protein changes that are protective and likely compensatory, such as proteasome components and chaperones (G); *n* = 3 (biological replicates), SEM, *\*p* < 0.05, *\*\*p* < 0.01, *\*\*\*p* < 0.001 (one-way ANOVA). See also S3 Data and S2 Fig. The data underlying the graphs shown in this figure can be found in the S6 Data file.

consistently regulated by muscle-specific eff RNAi and aging, whereas the remaining 302 proteins are discordantly regulated (Fig 7A–7D and S4 Data).

Proteins that were up-regulated by both eff RNAi and aging (the largest class, 514 proteins) were enriched for regulators of glutathione metabolism, chaperones, proteasome components, and vesicle transport (Fig 7B) and included Arc1 and Arc2. Commonly down-regulated proteins (186 proteins) included components of mitochondrial ribosomes and late endosomes, SNARE proteins, and proteases/peptidases (Fig 7C). The discordantly regulated proteins were enriched for oxidoreductases, mitochondrial and endoplasmic reticulum components, and secreted proteins (106 proteins that are up-regulated by aging but down-regulated by eff RNAi; Fig 7A), and regulators of cytoplasmic translation and mRNA splicing (196 proteins that are down-regulated by aging but up-regulated by eff RNAi; Fig 7D).

Similar results were also obtained with Venn diagrams (Fig 7E), which identified the proteins with overlapping, consistent, or discordant regulation by aging versus eff$^{RNAi}$. The threshold of $log_2FC > 0.3$ and $< -0.3$ was used as a criterion for the selection of significantly up- and down-regulated proteins for these analyses (Fig 7E). In this case, 23.9% of significantly regulated proteins (141) were up-regulated by both aging and eff$^{RNAi}$, 7.3% (43) were consistently down-regulated by both, and 3.2% (19) were oppositely regulated (Fig 7E).

Altogether, these analyses indicate that UBE2D/eff$^{RNAi}$ causes proteomic changes that recapitulate in part those induced by aging, indicating that UBE2D/eff is necessary to maintain a youthful proteome composition.

## Discussion

In this study, we have examined the role of the 21 *Drosophila* E2 ubiquitin-conjugating enzymes in modulating the aggregation of a model polyglutamine protein. By analyzing pathogenic huntingtin in the *Drosophila* retina, we found that E2s have diverse functions in regulating Htt-polyQ-GFP. Specifically, the knockdown of many E2s reduces polyglutamine protein aggregates (Fig 1), suggesting that ubiquitination by these E2s promotes sequestration of pathogenic huntingtin into aggresomes, as found in *C. elegans* [39]. Aggregation of Htt was previously found to decrease its toxicity [76,77], and therefore E2s that drive Htt aggregation may protect from polyglutamine disease via this mechanism. However, there were also RNAi interventions for other E2s, such as UBE2D/eff, and associated enzymes (DUBs and E3s) that increased the levels of high-molecular-weight assemblies of Htt-polyQ-GFP (Figs 1 and 2), suggesting that these E2s and interacting partners are necessary for Htt-polyQ-GFP degradation. Importantly, the E2s that affect Htt-polyQ-GFP aggregates may act via distinct or overlapping pathways, such as the modulation of proteolytic pathways (i.e., proteases, the autophagy/lysosome, and the ubiquitin/proteasome system) or by directly promoting the ubiquitination and degradation of Htt. However, IP-MS indicates that UBE2D/eff does not interact with Htt-polyQ-GFP (S5 Data), suggesting that Htt-polyQ-GFP might not be a direct ubiquitination target of UBE2D/eff.

In addition to this HD model, UBE2D/eff knockdown compromises protein quality control also in *Drosophila* skeletal muscle during aging (Figs 3 and 4). Specifically, UBE2D/eff protein levels decline with age in *Drosophila* skeletal muscle, and experimental knockdown of UBE2D/eff from a young age causes an accelerated buildup in insoluble poly-ubiquitinated proteins (Fig 3), which progressively accumulate during normal muscle aging. A previous report profiled the proteomics changes that occur during aging in the human vastus lateralis skeletal muscle [78]: interestingly, this data set reports UBE2D2 (a homolog of *Drosophila* eff) among the proteins that are significantly down-regulated with aging. Therefore, reduction in UBE2D levels and function could be an evolutionary conserved mechanism that contributes to the loss

**A** up with aging, down with effRNAi
106 proteins, 86 DAVID IDs

| enrichment score | |
|---|---|
| 2.39 | oxidoreductase activity |
| 2.17 | mitochondrion |
| 2.09 | valine, leucine degradation |
| 1.37 | iron, 4 sulfur cluster binding |
| 0.88 | endoplasmic reticulum |
| 0.45 | secreted |

**B** 514 proteins, 411 DAVID IDs
upregulated by both aging and effRNAi

| enrichment score | |
|---|---|
| 4.87 | glutathione transferase activity |
| 4.57 | unfolded protein binding |
| 3.81 | proteasome complex |
| 2.63 | nucleotide-binding |
| 2.58 | vesicle-mediated transport |
| 2.45 | UBA-like |
| 2.36 | UBX domain |
| 2.14 | imaginal disc growth factor |
| 1.94 | chaperone binding |
| 1.88 | innate immune response |

**C** 186 proteins, 146 DAVID IDs
downregulated by both aging and effRNAi

| enrichment score | |
|---|---|
| 5.54 | mitochondrial translation |
| 1.87 | Immunoglobulin domain |
| 1.71 | late endosome |
| 1.49 | CHK kinase-like |
| 1.34 | SNARE complex |
| 1.2 | carboxylesterase |
| 1.12 | pyridoxal phosphate binding |
| 0.83 | protein import into mitochondrial matrix |
| 0.78 | serine-type endopeptidase activity |

**D** down with aging, up with effRNAi
196 proteins, 150 DAVID IDs

| enrichment score | |
|---|---|
| 16.52 | cytoplasmic translation |
| 2.43 | chaperone |
| 1.99 | mRNA splicing, via spliceosome |
| 1.98 | actin binding |
| 1.46 | RNA splicing |
| 1.39 | nucleotide-binding |
| 1 | GTP-binding |

**Fig 7. Knockdown of the ubiquitin-conjugating enzyme UBE2D/eff drives proteomic changes associated with aging in *Drosophila* skeletal muscle. (A–D)** Cross-comparison of TMT mass spectrometry data from *Drosophila* skeletal muscle identifies substantial overlap in the proteomic changes induced by UBE2D/eff knockdown in young age compared to the changes that are induced by aging. The *x*-axis displays the significant ($p < 0.05$) changes ($\log_2$FC) induced in skeletal muscle by eff RNAi (*Mhc>eff$^{RNAi}$*) compared to control mcherry RNAi (*Mhc>mcherry$^{RNAi}$*) at 2

weeks of age ($n$ = 3 biological replicates/group). The $y$-axis reports the significant ($p < 0.05$) changes induced by aging in the skeletal muscle of control flies ($w^{1118}$) when comparing 8 weeks (old) vs. 1 week (young), with $n$ = 5 biological replicates/group. Among the proteins that are significantly regulated ($p < 0.05$) by both UBE2D/eff knockdown and aging, approximately 70% are consistently regulated, i.e., either up-regulated (B, 51%) or down-regulated (A, 18.5%) by both, whereas the remaining approximately 30% is regulated oppositely by UBE2D/eff RNAi versus aging (A, D). The protein categories that are overrepresented in each group are indicated in (A–D) alongside the enrichment score. Representative proteins that are significantly regulated by UBE2D/eff RNAi in a consistent or discordant manner are shown in the graph and include Arc1 and Arc2. (E) Venn diagrams representing the overlap in the regulation of protein levels by aging and eff$^{RNAi}$. These graphs were obtained from the list of significantly regulated proteins modulated by aging and eff$^{RNAi}$ (S4 Data). The threshold of log$_2$FC>0.3 and <-0.3 was further applied for selecting up- and down-regulated proteins. The data underlying the graphs shown in this figure can be found in the S6 Data file.

of muscle proteostasis, which is an important component of aging in both *Drosophila* and humans [25,48–51,54,79–90]. Interestingly, transgenic expression of human UBE2D2 in *Drosophila* partially rescues the proteostasis deficits caused by eff$^{RNAi}$ (Figs 2, 4, and 5), and this occurs by re-establishing the physiological levels of eff$^{RNAi}$-regulated proteins (Fig 6), further reinforcing the evolutionary conservation and importance of UBE2D for muscle protein quality control.

Previously, UBE2D ubiquitin-conjugating enzymes (UBE2D1/2/3/4) were found to contribute to several cellular processes that impact proteostasis. These include the association of UBE2Ds with the E3 ubiquitin ligase CHIP to promote the degradation of misfolded proteins [27]. Likewise, UBE2Ds were found to confer resistance to elevated temperatures in yeast by promoting the selective ubiquitination and degradation of misfolded proteins [31,91]. Moreover, UBE2Ds can interact with the E3 ligase parkin to promote mitophagy [29] and with the E3 ligase RNF138 to mediate DNA repair [92]. In addition to these cellular functions, UBE2D also plays a key role in protein import into peroxisomes [35,93–96].

Here, we found that UBE2D/eff is an E2 ubiquitin-conjugating enzyme that is key for maintaining protein quality control during aging (Figs 1–6). Considering that UBE2D/eff is one out of 21 different E2 ubiquitin-conjugating enzymes in *Drosophila*, it is unlikely that UBE2D/eff RNAi alone reduces the total ubiquitination capacity of the cell. In fact, UBE2D/eff knockdown paradoxically increases the levels of insoluble poly-ubiquitinated proteins (Figs 3 and 4), which likely consist of proteins different from UBE2D/eff ubiquitination substrates. How UBE2D/eff knockdown deranges proteostasis and promotes the accumulation of poly-ubiquitinated proteins may occur via multiple mechanisms, including those listed above and previously found in other contexts. For example, UBE2D/eff knockdown may derange proteostasis because of the previously reported roles of UBE2D/eff in promoting mitophagy [29] and in the removal of unfolded proteins [31,91] in concert with the E3 ubiquitin ligase CHIP [27]. Accumulation of such misfolded proteins may clog or burden the proteasome and hence generally impair the degradation of poly-ubiquitinated proteins, which would in turn accumulate, as we have found to happen in muscles with eff$^{RNAi}$ (Figs 3 and 4). Consistent with this model, several chaperones are up-regulated in eff$^{RNAi}$ muscles (Figs 6 and 7), possibly indicative of the induction of an unfolded protein response.

In addition to these previously known roles of UBE2D in proteostasis, we have now found that UBE2D/eff knockdown reduces the proteolytic activity of the proteasome in skeletal muscle and that these defects can be rescued by expression of human UBE2D2 (i.e., one of its human homologs): this indicates that UBE2D/eff is necessary to maintain proteasomal activity (Fig 5), which is key for proteostasis in all tissues, including skeletal muscles [49,97,98]. The capacity of UBE2D/eff to modulate proteasome activity may therefore explain the role of UBE2D/eff in proteostasis during skeletal muscle aging (Figs 3–7) as well as in the context of Huntington's disease (Figs 1 and 2). Altogether, these findings contribute to the growing

understanding of the role of E2 enzymes in modulating proteasome-mediated proteolysis. In this respect, in addition to participating in target protein ubiquitination, a few E2 and E3 enzymes were previously found to associate with the proteasome: these proteasome-interacting proteins modulate proteasome function and increase the resistance to homeostatic challenges [26,32,34,60]. For instance, the ubiquitin ligase UBR4 interacts with several proteasome components and this association increases in response to proteasome inhibition with MG132 [25,26]. This interaction is functionally important because UBR4 knockout reduces proteasome activity in mouse skeletal muscle [25]. In this framework, it is interesting to note that UBE2D/eff is homologous to yeast UBC4 and UBC5: these E2 enzymes bind to the proteasome, and this association further increases with heat stress [31], a condition where proteasome function is modulated by ubiquitination of proteasome components [34]. Further studies in mammalian systems found that the ligase CHIP/STUB1 poly-ubiquitinates and sequesters proteasomes in insoluble aggregates upon stress [62]. Interestingly, UBE2D enzymes were previously found to work in concert with CHIP/STUB1 [27], and therefore UBE2D/eff may modulate proteasome activity via CHIP/STUB1 and other interacting E3 ligases [35].

While we have found that UBE2D is required for the optimal proteolytic activity of the proteasome, additional mechanisms may also play a role in the preservation of protein quality control by UBE2D/eff. For example, proteostasis and survival might be reduced by UBE2D/eff knockdown because of the accumulation of specific UBE2D protein targets that directly perturb these processes. While many substrates of UBE2D/eff ubiquitination accumulate and may play such role, Arc1 and the related Arc2 protein are of particular interest. Specifically, Arc1 and Arc2 protein levels increase upon UBE2D/eff knockdown and, conversely, they are reduced by hUBE2D2. Interestingly, Arc1 up-regulation was found to be cytotoxic and to drive neurodegeneration in a *Drosophila* Alzheimer's disease model [70]. This suggests that the accumulation of Arc1/2 in response to UBE2D/eff knockdown may contribute to derange proteostasis and reduce survival. Arc1 and Arc2 protein levels also increase in response to aging in skeletal muscle (Fig 7). More globally, eff RNAi induces many proteomic changes reminiscent of aging: approximately 70% of the proteins that are significantly modulated by eff RNAi in young age are also consistently regulated by aging (Fig 7). These findings, therefore, indicate that UBE2D/eff maintains a youthful proteome and that the age-dependent decline in UBE2D/eff protein levels (Fig 3), which is also found in humans [78], may contribute to the proteomic changes that occur with aging. However, some of these proteomic changes may derive from a transcriptional stress response triggered by the eff$^{RNAi}$-induced decline in proteasome activity. In support of this hypothesis, perturbation of proteasome function and knockdown of proteasome components were previously found to induce the compensatory up-regulation of heat shock proteins and proteasome components [13,71,73,99–102]. A similar transcriptional stress response is also induced during normal aging in skeletal muscle [49,103]. Therefore, up-regulation of the mRNA and protein levels of chaperones and the proteasome maturation factor Pomp [71,72] may indicate the induction of the proteasome stress response (i.e., an unfolded protein response) upon UBE2D/eff knockdown and aging.

In addition to reshaping the composition of the muscle proteome and impairing proteostasis, we have found that muscle-specific UBE2D/eff knockdown reduces the organism's lifespan, and that this is partially rescued by concomitant expression of an RNAi-resistant human UBE2D2 transgene which re-establishes the physiological levels of several UBE2D/eff-regulated targets such as Arc1/2. These findings, therefore, reinforce the notion that skeletal muscle is a key tissue that influences systemic aging, and that preserving muscle protein quality control is necessary for optimal organismal survival [25,48,58,104–107].

In summary, our study highlights the important role of the ubiquitin-conjugating enzyme UBE2D/eff in maintaining protein quality control during aging (Fig 8), suggesting that

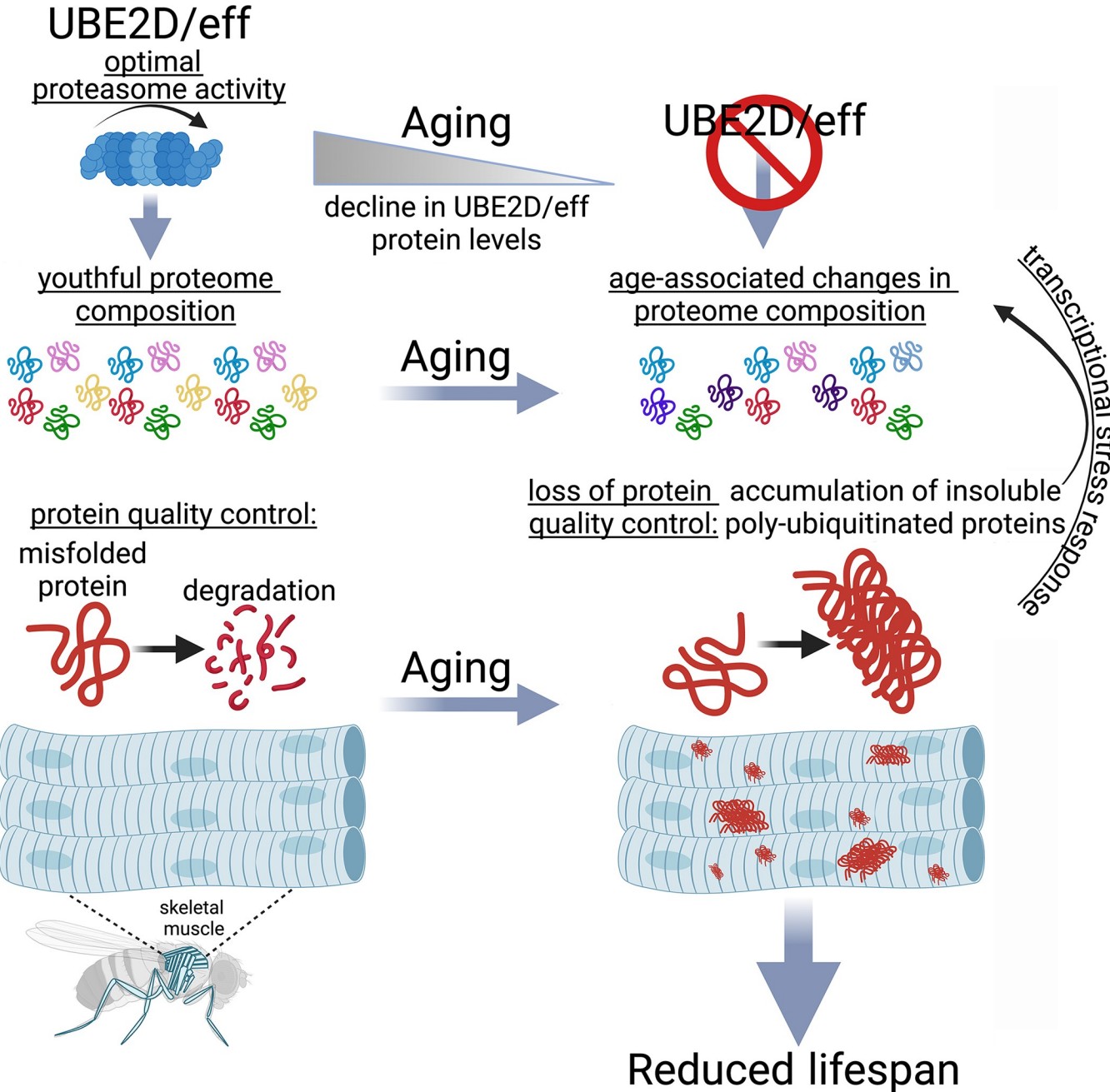

**Fig 8. The E2 enzyme UBE2D/eff preserves proteostasis and maintains a youthful proteome composition in skeletal muscle during aging by sustaining the proteolytic activity of the proteasome.** The ubiquitin-conjugating enzyme UBE2D/eff has a key role in proteostasis in skeletal muscle by maintaining optimal proteasome activity. UBE2D/eff protein levels decline during aging. Experimentally reducing UBE2D/eff levels from a young age causes a loss in protein quality control, a consequent precocious surge in the levels of insoluble poly-ubiquitinated proteins (which normally accumulate only in old age because of age-associated defects in proteostasis), and shortens lifespan. Proteomics surveys indicate that UBE2D/eff knockdown induces proteomic changes similar to aging, and that such UBE2D/eff[RNAi]-induced changes are rescued by transgenic expression of its human homolog UBE2D2. Some of the proteins that are up-regulated by aging and UBE2D/eff[RNAi] include proteostasis regulators (such as chaperones and Pomp) that are transcriptionally induced as part of an adaptive stress response to the decline in proteostasis. Altogether, these findings indicate that UBE2D/eff is necessary to maintain a youthful proteome and to ensure muscle protein quality control during aging by sustaining proteasome activity.

interventions that promote UBE2D function may contrast skeletal muscle aging and other age-related diseases that arise from the loss of proteostasis.

## Materials and methods

### *Drosophila* husbandry and stocks

Flies were kept (approximately 25 flies/tube) at 25˚C, 60% humidity, and a 12 h/12 h light-dark cycle in tubes containing cornmeal/soy flour/yeast fly food. Survival analysis was done at 25˚C and the fly food was changed every 2 to 3 days [108]. All experiments were done with male flies. Fly stocks were obtained from the Bloomington *Drosophila* Stock Center (BDSC), the Vienna *Drosophila* Resource Center (VDRC), and the National Institute of Genetics (NIG-Fly, Japan). The following fly stocks were utilized: *MhcF3-Gal4* ([25,109] BL#38464), *Mhc-Gal4* [48], control *UAS-white*[RNAi] (BL#33623 and v30033), control *UAS-luciferase*[RNAi] (BL#31603), *UAS-eff*[RNAi] (BL#35431, BL#31875, #7425R-2, and v26012), *UAS-mcherry* (BL#35787), *UAS-hUBE2D2* (BL#76819), and *UAS-mcherry*[RNAi] (BL#35785). The eff[RNAi] stock used is BL#35431 (the strongest line, S2 Fig) unless otherwise noted. *GMR-Gal4* and *UAS-Htt-72Q-GFP* flies were previously described [41,110]. The fly stocks used for the screen in Fig 1 are reported in S1 Data.

### Whole-mount immunostaining of *Drosophila* skeletal muscle

The immunostaining of flight skeletal muscle was done as previously described [48,111,112]. In brief, thoraces were dissected, fixed for 30 min in PBS with 4% paraformaldehyde and 0.1% Triton X-100 at room temperature, washed >3 times in PBS with 0.1% Triton X-100 at room temperature, and immunostained overnight at 4˚C with rabbit anti-poly-ubiquitin (FK2; Enzo Life Sciences #BML-PW8810-0100). After washes with PBS with 0.1% Triton X-100, the samples were incubated with secondary antibodies and Alexa635-phalloidin for 2 h at room temperature, washed, and mounted in an antifade medium. Subsequently, the samples were imaged on a Nikon C2 confocal microscope. Image analysis was done with NIH ImageJ.

### Analysis of pathogenic Huntingtin aggregation

Pathogenic huntingtin-polyQ72-GFP protein aggregates [41] were imaged with an epifluorescence ZEISS SteREO Discovery V12 microscope with consistent exposure time and settings. The acquired grayscale images were then analyzed in an automated manner by using Cell Profiler 3.0.0 (cellprofiler.org) to determine the number and/or total area of protein aggregates (Huntingtin-polyQ72-GFP speckles) normalized by the retinal tissue area. Background fluorescence not corresponding to Huntingtin-polyQ72-GFP aggregates was excluded by thresholding. This analysis was done with male flies after aging at 29˚C for 30 days; the same conditions were applied to all the samples and the respective controls in any given experiment. The raw values for each set of flies were normalized by the corresponding raw values of the negative control RNAi intervention from the same fly stock collection. In the graphs in Fig 1B and 1C, the relative fold changes compared to control RNAi interventions are shown. Each data point in these graphs represents a single RNAi targeting the corresponding gene. For each RNAi, the mean of 5 biological replicates is shown, with each biological replicate representing a single eye (each from a distinct animal). The variability in the analysis of each single gene is due to the distinct efficacy of the different RNAi lines targeting the same gene.

### Western blot analysis of flies that express Hungtingtin-polyQ72-GFP

For the western blot analysis of flies that express Hungtingtin-polyQ72-GFP, 30-day-old flies (*n* = 5/sample) were homogenized in RIPA buffer containing 8M urea and 5% SDS, and

centrifuged at 14,000 rpm for 10 min at 4˚C. The supernatants were collected and analyzed on 4% to 20% SDS-PAGE with anti-GFP antibodies (Cell Signaling Technologies D5.1, #2956). Anti-α-tubulin antibodies (Cell Signaling Technologies, #2125) were used as loading controls. The IP-MS of huntingtin-polyQ72-GFP was done according to standard procedures with GFP-trap agarose (Ptglab, #gta) [35].

## Western blot analysis of detergent-soluble and -insoluble fractions

Western blots for detergent-soluble and insoluble fractions were done as before [48,57,80,110]. In brief, thoraces were dissected from 20 male flies/sample and homogenized in ice-cold PBS with 1% Triton X-100 containing protease and phosphatase inhibitors. Homogenates were centrifuged at 14,000 rpm at 4˚C and the supernatants were collected (Triton X-100 soluble fraction). The remaining pellet was washed in ice-cold PBS with 1% Triton X-100. The pellet was then resuspended in RIPA buffer containing 8M urea and 5% SDS, centrifuged at 14,000 rpm at 4˚C, and the supernatant (Triton X-100 insoluble fraction) was collected. The detergent-soluble and insoluble fractions were then analyzed on 4% to 20% SDS-PAGE with the following primary antibodies: anti-ubiquitin (Cell Signaling Technologies P4D1, #3936), anti-Ref (2)P (Abcam, #178840), anti-α-tubulin (Cell Signaling Technologies, #2125), anti-26S proteasome α (Santa Cruz, IIG7, #sc-65755; which detects α subunits of the 20S catalytic core of the *Drosophila* proteasome), anti-26S proteasome p42A/Rpn7 (Santa Cruz, 123, #sc-65750; which detects the p42A subunit of the 19S regulatory lid complex of the *Drosophila* proteasome), and anti-Atg8/GABARAP (abcam, #ab109364).

## Proteasome activity assays

Luminescence assays to monitor proteasome activity were done as previously described [25,35,59], with minor modifications. Specifically, 10 thoraces were collected and homogenized in 100 µl of PBS with a Next Advance bullet blender and 0.5-mm zirconium beads. After homogenization, 100 µl of PBS was added and the samples were vortexed briefly. The homogenate was collected after centrifugation at high speed for 10 min. The homogenates from 3 samples prepared as described above were combined to form a biological replicate, each having a total concentration of 30 thoraces per 600 µl of PBS. Caspase-like, chymotrypsin-like, and trypsin-like proteolytic activities of the proteasome were estimated with the Proteasome-Glo 3-substrate assay system (Promega, #G8531). For this assay, 25 µl of tissue homogenate was added to each well of a 96-well white solid plate (Corning); 25 µl of the Proteasome-Glo reagent was added to each well, and the plate was then mixed on a plate shaker at room temperature, shielded from light. After 30 min, the luminescence was measured with a Tecan Infinite 200 Pro plate reader. Three technical replicates were examined for each biological replicate.

## qRT-PCR

qRT-PCR was performed as previously described [26,113–117]. Total RNA was extracted with the TRIzol reagent (Life Technologies) from *Drosophila* thoraces, consisting primarily of skeletal muscle, from >20 male flies/replicate, followed by reverse transcription with the iScript cDNA synthesis kit (Bio-Rad). qRT-PCR was performed with SYBR Green and a CFX96 apparatus (Bio-Rad). Three biological replicates were used for each genotype and time point. Alpha-tubulin at 84B (*Tub84B*) was used as a normalization reference. Whole flies at 1 week of age were utilized for the qRT-PCR in Fig 1D to detect *Htt* and *GFP* levels. The comparative $C_T$ method was used for the relative quantitation of mRNA levels. The following qRT-PCR oligos were used:

*eff*: 5′-CAATAATGGGCCCGCCGGA-3′ and 5′-GCGCGTTGTAAAAGCCACTT-3′
*Ref(2)P*: 5′-CCCGTCTACCACACTGATGA-3′ and 5′-TTACGTCCAAGGCAGCTGAG-3′
*Atg8a*: 5′-CCGAGGATGCCCTCTTCTTC-3′ and 5′-CCGACCGGAGCAAAGTTAGT-3′
*Pomp*: 5′-CAACCGCAACATGCAGATGC-3′ and 5′-CTGGACGAAAGGAAGGGCAG-3′
*Hsp22*: 5′-GGACGTCAAGGACTACAGCG-3′ and 5′-GGAGGATTGGGCACACTGAT-3′
*Hsp26*: 5′-CCCCATCTACGAGCTTGGAC-3′ and 5′-TGGAATCCATCCTTGCCCAC-3′
*Hsp70*: 5′-CCAAGATGCATCAGCAGGGA-3′ and 5′-TTGGCTTTAGTCGACCTCCT-3′
*Hsp83*: 5′-CGAAAAACATACATACAAGATGCCA-3′ and 5′-AGCCTGGAATG-
CAAAGGTCT-3′
*GFP*: 5′-ACGTAAACGGCCACAAGTTC-3′ and 5′-CTTCATGTGGTCGGGGTAGC-3′
*Htt (exon 1)*: 5′-CCTGGATCCCTGGTGAGCAA-3′ and 5′-
GCTGAACTTGTGGCCGTTTA-3′
*Tub84B*: 5′-GTTTGTCAAGCCTCATAGCCG-3′ and 5′-GGAAGTGTTTCACACGC-
GAC-3′

## Protein sample preparation, protein digestion, and peptide isobaric labeling by tandem mass tags

For each TMT *Drosophila* sample, 50 thoraces (consisting primarily of skeletal muscle) from 1-week-old and 8-week-old male $w^{1118}$ flies (Fig 3) and from 2-week-old male flies of the genotypes indicated (Fig 6) were collected and homogenized in 8 M urea lysis buffer (50 mM HEPES, pH 8.5, 8 M urea) [118–120]. After homogenization with 0.5-mm zirconium beads in a Next Advance bullet blender, 0.5% sodium deoxycholate was added to the tissue homogenates, which were then pelleted to remove fly debris. The resulting supernatant was submitted for TMT mass spectrometry and digested with LysC (Wako) at an enzyme-to-substrate ratio of 1:100 (w/w) for 2 h in the presence of 1 mM DTT. Following this, the samples were diluted to a final 2 M Urea concentration with 50 mM HEPES (pH 8.5), and further digested with trypsin (Promega) at an enzyme-to-substrate ratio of 1:50 (w/w) for at least 3 h. The peptides were reduced by adding 1 mM DTT for 30 min at room temperature (RT) followed by alkylation with 10 mM iodoacetamide (IAA) for 30 min in the dark at RT. The unreacted IAA was quenched with 30 mM DTT for 30 min. Finally, the digestion was terminated and acidified by adding trifluoroacetic acid (TFA) to 1%, desalted using C18 cartridges (Harvard Apparatus), and dried by speed vac. The purified peptides were resuspended in 50 mM HEPES (pH 8.5) and labeled with 16-plex Tandem Mass Tag (TMT) reagents (Thermo Scientific) following the manufacturer's recommendations and our optimized protocol [56].

## Two-dimensional HPLC and mass spectrometry

The TMT-labeled samples were mixed equally, desalted, and fractionated on an offline HPLC (Agilent 1220) by using basic pH reverse-phase liquid chromatography (pH 8.0, XBridge C18 column, 4.6 mm × 25 cm, 3.5 μm particle size, Waters). The fractions were dried and resuspended in 5% formic acid and analyzed by acidic pH reverse phase LC-MS/MS analysis. The peptide samples were loaded on a nanoscale capillary reverse phase C18 column (New objective, 75 μm ID × ~25 cm, 1.9 μm C18 resin from Dr. Maisch GmbH) by an HPLC system (Thermo Ultimate 3000) and eluted by a 90-min gradient. The eluted peptides were ionized by electrospray ionization and detected by an inline Orbitrap Fusion mass spectrometer (Thermo Scientific). The mass spectrometer was operated in data-dependent mode with a survey scan in Orbitrap (60,000 resolution, $1 \times 10^6$ AGC target and 50 ms maximal ion time) and MS/MS high-resolution scans (60,000 resolution, $2 \times 10^5$ AGC target, 120 ms maximal ion time, 32 HCD normalized collision energy, 1 *m/z* isolation window, and 15 s dynamic exclusion).

## MS data analysis

The MS/MS raw files were processed by the tag-based hybrid search engine JUMP [121]. The raw data were searched against the UniProt *Drosophila* database concatenated with a reversed decoy database for evaluating false discovery rates. Searches were performed by using a 15-ppm mass tolerance for both precursor and product ions, fully tryptic restriction with 2 maximal missed cleavages, 3 maximal modification sites, and the assignment of *a*, *b*, and *y* ions. TMT tags on Lys and N-termini (+304.20715 Da) were used for static modifications and Met oxidation (+15.99492 Da) was considered as a dynamic modification. Matched MS/MS spectra were filtered by mass accuracy and matching scores to reduce the protein false discovery rate to approximately 1%. Proteins were quantified by summing reporter ion intensities across all matched PSMs using the JUMP software suite [122]. Categories enriched in protein sets were identified with DAVID [123]. Protein homology was determined with DIOPT (https://www.flyrnai.org/diopt).

## Statistical analysis

All experiments were performed with biological triplicates unless otherwise indicated. The unpaired two-tailed Student's *t* test was used to compare the means of 2 independent groups to each other. One-way ANOVA with post hoc testing was used for multiple comparisons of more than 2 groups of normally distributed data. Survival data was analyzed with OASIS 2 [124] by using log-rank tests. The "*n*" for each experiment can be found in the figures and/or figure legends and represents independently generated samples, including individual flies for lifespan assays, and batches of flies or fly thoraces for other assays. Bar graphs represent the mean ± SEM or ±SD as indicated in the figure legend. Throughout the figures, asterisks indicate a significant *p*-value (*$p < 0.05$). Statistical analyses were done with Excel and GraphPad Prism.

## Supporting information

**S1 Fig. Knockdown of some E2 ubiquitin-conjugating enzymes impacts the transgenic expression of huntingtin-polyQ72-GFP.** qRT-PCR from the heads of flies that express *huntingtin-polyQ72-GFP* in the retina (with *GMR-Gal4*) alongside a control or RNAi for some of the E2s that scored in the Htt screen (Fig 1): Ubc4, Bruce, and CG5402. This gene expression analysis indicates that there is a substantial decline in the transgenic expression of huntingtin-polyQ72-GFP (as indicated by the *GFP* and *Htt* mRNA levels) upon knockdown of Ubc4 and Bruce, whereas there is a partial increase in response to CG5402 RNAi. The graphs display the mean ± SEM with *n* = 3 (biological replicates). Statistical analysis was done with one-way ANOVA and Dunnett's multiple comparisons test; *$p < 0.05$. These findings indicate that the knockdown of Ubc4, Bruce, and CG5402 may modulate the amount of GFP-tagged huntingtin-polyQ aggregates via changes in the expression of the *Htt-polyQ72-GFP* transgene: on this basis, these E2s were not considered for further analyses. Mechanistically, these E2s may modulate transgenic expression by regulating histone ubiquitination and degradation (which generally alters transcriptional activity), as previously shown for the Ubc4 ortholog UBE2K (PMID: **32451438**). Related to Fig 1. The data underlying the graphs shown in this figure can be found in the S6 Data file.
(TIFF)

**S2 Fig. UBE2D/eff knockdown induces the expression of proteostasis regulators.** qRT-PCR analysis of skeletal muscle with eff^RNAi and control^RNAi. Knockdown of eff up-regulates the expression of the proteasome assembly factor *Pomp* and of the chaperones *Hsp22*, *Hsp26*, and *Hsp70*. The graphs display the mean ± SEM with *n* = 3 (biological replicates). Statistical analysis was done with one-way ANOVA and Dunnett's multiple comparisons test; *$p < 0.05$;

**p < 0.01; ***p < 0.001. These findings suggest that a stress response (presumably related to the unfolded protein response or the proteasome stress response) is triggered by the decline in protein quality control due to UBE2D/eff knockdown. Interestingly, this response is primarily induced only by the stronger eff$^{RNAi}$ line (BL#35431), suggesting that there could be a threshold of eff knockdown that is required for the transcriptional induction of this response. The data underlying the graphs shown in this figure can be found in the S6 Data file.
(TIF)

**S1 Data. RNAi screen for E2 ubiquitin-conjugating enzymes and associated E3 ubiquitin ligases that regulate pathogenic huntingtin-polyQ72-GFP protein aggregates in *Drosophila*.** Each E2/E3 was analyzed with multiple RNAi lines and *n* = 5 biological replicates were examined for each RNAi line. Related to Fig 1.
(XLSX)

**S2 Data. Mass spectrometry analysis of age-induced protein changes in *Drosophila* skeletal muscle. Tandem mass tag (TMT) mass spectrometry identifies age-related protein changes in the skeletal muscle (thoraces) of old vs. young *w$^{1118}$* male flies (8 vs. 1 week old).** A tab also reports the list of age-regulated deubiquitinating enzymes (DUBs), E1s, E2s, and E3 enzymes. Related to Figs 3 and 7.
(XLSX)

**S3 Data. Mass spectrometry analysis of protein changes induced by UBE2D/eff knockdown in *Drosophila* skeletal muscle, and their rescue by co-expression of human UBE2D2.** Tandem mass tag (TMT) mass spectrometry identifies protein changes induced in skeletal muscle by eff RNAi (compared to control interventions) and their rescue by co-expression of hUBE2D2 (the human homolog of eff). A dedicated tab reports the modulation of proteasome levels by UBE2D/eff$^{RNAi}$. Related to Figs 5 and 6.
(XLSX)

**S4 Data. Cross-comparison of proteomic changes induced by aging vs. eff RNAi in skeletal muscle.** Cross-comparison of the mass spectrometry data from S2 and S3 Data files. Related to Fig 7.
(XLSX)

**S5 Data. IP-MS of proteins that interact with Htt-polyQ72-GFP. IP-MS was utilized to define the interactome of huntingtin-polyQ72-GFP, compared to the model proteasome substrate GFP-CL1 (a GFP-tagged degron-containing protein).** UBE2D/eff is not detected among the Htt-interacting proteins, suggesting that Htt-polyQ72-GFP might not be a direct ubiquitination target of UBE2D/eff. Related to Figs 1 and 2.
(XLSX)

**S6 Data. Source data.** Related to Figs 1–7 and S1–S2.
(XLSX)

**S1 Raw Images. Uncropped full scans of western blots corresponding to Figs 1E, 2C, 3E, 4A, 5D, and 5E.**
(PDF)

## Acknowledgments

We thank the Light Microscopy facility at St. Jude Children's Research Hospital. *Drosophila* stocks were provided by the VDRC, NIG-Fly, and the Bloomington stock centers. The scheme was drawn with BioRender.

## Author Contributions

**Conceptualization:** Liam C. Hunt, Fabio Demontis.

**Data curation:** Liam C. Hunt, Michelle Curley, Kudzai Nyamkondiwa, Anna Stephan, Kanisha Kavdia, Vishwajeeth R. Pagala.

**Formal analysis:** Liam C. Hunt, Michelle Curley, Kanisha Kavdia, Vishwajeeth R. Pagala.

**Funding acquisition:** Fabio Demontis.

**Investigation:** Liam C. Hunt, Michelle Curley, Kudzai Nyamkondiwa, Jianqin Jiao, Kanisha Kavdia, Vishwajeeth R. Pagala.

**Methodology:** Liam C. Hunt, Michelle Curley, Vishwajeeth R. Pagala, Junmin Peng.

**Project administration:** Fabio Demontis.

**Supervision:** Junmin Peng, Fabio Demontis.

**Validation:** Liam C. Hunt.

**Visualization:** Liam C. Hunt, Michelle Curley.

**Writing – original draft:** Fabio Demontis.

**Writing – review & editing:** Liam C. Hunt, Michelle Curley, Anna Stephan, Junmin Peng, Fabio Demontis.

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
