## [Editor Report · Decision Letter 0]

17 May 2024

Dear Dr Demontis, 

Thank you for submitting your manuscript entitled "The ubiquitin-conjugating enzyme UBE2D/eff maintains a youthful proteome and ensures protein quality control during aging" for consideration as a Research Article by PLOS Biology.

Your manuscript has now been evaluated by the PLOS Biology editorial staff as well as by an academic editor with relevant expertise and I am writing to let you know that we would like to send your submission out for external peer review.

Once your full submission is complete, your paper will undergo a series of checks in preparation for peer review. After your manuscript has passed the checks it will be sent out for review. To provide the metadata for your submission, please Login to Editorial Manager (https://www.editorialmanager.com/pbiology) within two working days, i.e. by May 21 2024 11:59PM.

Kind regards,

Ines

--

Ines Alvarez-Garcia, PhD

Senior Editor

PLOS Biology

---

## [Decision Letter · Decision Letter 1]

2 Aug 2024

Dear Dr Demontis,

Thank you for your patience while your manuscript entitled "The ubiquitin-conjugating enzyme UBE2D/eff maintains a youthful proteome and ensures protein quality control during aging" was peer-reviewed at PLOS Biology. Please also accept my apologies for the delay in sending you our decision. The manuscript has now been evaluated by the PLOS Biology editors, an Academic Editor with relevant expertise, and by two independent reviewers. 

The reviews are attached below. As you will see, the reviewers find the conclusions interesting and significant for the field. Nevertheless, they raise several points that would need to be addressed before considering the manuscript for publication. Reviewer 1 thinks the significance of the study could increase if it is shown whether eff E2 enzyme functions, and whether it has an effect on aggregate turnover, proteasome function or any other main step during protein turnover. This reviewer also thinks you should measure proteasome activity to test if eff affects its function, to validate the proteomics candidates. Reviewer 2 suggests testing the efficacy of different RNAi lines and checking if the difference between the eff-RNAi-induce proteins tested also occurs at the mRNA level.

In light of the reviews, we would like to invite you to revise the work to thoroughly address the reviewers' reports. Given the extent of revision needed, we cannot make a decision about publication until we have seen the revised manuscript and your response to the reviewers' comments. Your revised manuscript is likely to be sent for further evaluation by all or a subset of the reviewers.

**IMPORTANT - SUBMITTING YOUR REVISION**

3. Resubmission Checklist

a) *PLOS Data Policy*

b) *Published Peer Review*

d) *Blurb*

Please also provide a blurb which (if accepted) will be included in our weekly and monthly Electronic Table of Contents, sent out to readers of PLOS Biology, and may be used to promote your article in social media. The blurb should be about 30-40 words long and is subject to editorial changes. It should, without exaggeration, entice people to read your manuscript. It should not be redundant with the title and should not contain acronyms or abbreviations. For examples, view our author guidelines: https://journals.plos.org/plosbiology/s/revising-your-manuscript#loc-blurb

Sincerely,

Ines

--

Ines Alvarez-Garcia, PhD

Senior Editor

PLOS Biology

Reviewers' comments

Rev. 1:

Overall Recommendation: Minor Revisions. This is an interesting paper with good experimental data. Minor experimental and a few editorial additions will make it suitable for publication in PLOS

Biology.

Novelty: This is an exciting line of investigation, and the hypothesis and experimental plan is novel.

Significance is Moderate. The significance of this work could be increased if it can be shown where eff E2 enzyme is functions, and whether it has a general effect of overall aggregate turnover, or more specifically on proteasome function, or some other key step in protein turnover.

Technical merits: The experimental approach is sound, and the results are convincing. However, some improvements are recommended. Measurement of proteasome activity will determine if eff affects the function of this multicatalytic protease. Measurement of proteasome assembly will similarly test this. These experiments are important for completion of this solid body of work, and can be performed simultaneously in a native gel. The authors should explain why they selected eff for further investigation, rather than Ubc6, which seemed to have a stronger effect.

Statistics: Adequate, although the authors should indicate on the figures what regions of the gels were quantified.

Supplementary Results: The proteomic studies are not convincing because none of it is experimentally verified. Moreover, the nature of the changes in protein levels could have many explanations, none of them related directly to eff expression.

General comments:

The general approach to use Htt-polyQ to mimic the effect of E2 enzymes on protein aggregates is

valid. To extrapolate that the turnover of Htt-polyQ by Eff is related to aging specifically, and not

generally to cellular homeostasis is less convincing.

RNAi of certain E2’s resulted in reduced amount of Htt aggregates, suggesting they contribute to the formation rather than degradation of these protein complexes. Is there a cumulative effective of knocking down multiple E2’s that contribute to this phenomenon? Are deletion mutants available, and have they been examined for Htt aggregate formation?

The authors identified four E2’s that promoted Htt degradation. Among these eff was examined further. Since RNAi of eff alone could reduce the turnover of Htt, it appears that its function is not

fully overlapping with Ubc6, Ubc10, and Ubc84D. For instance, there may be multiple targets for ubiquitination in the path for Htt-polyQ degradation, including the aggregated protein itself, but also

the proteasome, and other regulatory components. What is the effect of RNAi of multiple E2’s simultaneously? Do the aggregates appear earlier, or are there higher amounts? If not, how does one explain their collective effect on Htt degradation?

The rescue of the Htt degradation defect caused by RNAi of either effort Ubc6 by human homologs is exciting and convincing.

One consideration that the authors should address is whether the turnover of other normally degraded proteins is reduced when eff (or Ubc6) expression is blocked by RNAi. Aggregated proteins, such as Htt-polyQ, may form unproductive interactions with the proteasome, and thereby interfere with the degradation of other cellular proteins. This indirect effect on proteasome function could cause aging-specific effects. In the fly model used in this work, is there any evidence that proteasomes co-localize with the Htt-GFP aggregates? It is important to establish that proteasome activity is unaffected, especially since ubiquitination of certain proteasome

subunits can affect catalytic activity. This is a straightforward assay that can be measured using readily available fluorogenic substrates. Amore general effect of RNAi against eff is suggested by the increased levels of total polyUb proteins in both soluble and insoluble fractions of skeletal muscle cells.

If eff or Ubc6 regulate proteasome activity by ubiquitination, the results would be similar, but would not be due to a failure to ubiqutinate Htt-polyQ.

The authors should determine if the fluorescence generated by Htt-GFP is altered once it is present in an aggregate. If fluorescence is increased or decreased, it would affect the interpretation of their results. Reflecting on the work of Johnston and Kopito on the dynamic presence of CFTR-GFP aggregates in aggresomes, they observed that this process is reversible. It would therefore be useful to know if Htt-GFP can also be released from aggregates (and degraded) once eff expression is restored.

The authors do not show that Htt is the target of eff: only that there is a correlation between eff

levels and Htt presence in aggregates.

Whereas the expression of human UBE2D2 rescued the retinal defect caused by RNAi shut down of eff expression, a similar rescue of insoluble aggregates in muscle tissue was not observed. Since the authors presume that eff reduces the levels of protein aggregates, shouldn’t they have seen a decrease in insoluble aggregates with the expression of human UBE2D2? Does the prolongation of lifespan by expression of human UB2D2 indicate that only soluble aggregates are potentially toxic and result in aging?

Analysis of eff-induced protein changes revealed many house keeping proteins with altered expression (Fig. 5F). Whether they represent genuine targets of eff is unclear. Changes in these protein levels could simply reflect altered proteasome function arising from altered levels of aggregated Htt-polyQ or other aggregated proteins. These mass spec proteomic studies are unconvincing and not informative because none of the proteomic findings is experimentally validated.

Based on the elevated levels of Pomp assembly factor the authors should investigate if proteasome assembly is altered. This simple test will address two questions. 1. Is proteasome activity altered, and 2, is proteasome composition altered? Simple experiment.

The observation by other investigators that UBE2D interacts with the CHIP E3 ligase in the turnover of misfolded proteins, and heat stress in yeast, is strongly consistent with the work described here. The authors recognize the cellular targets other than Htt may be the prime contributors to aggregate-induced aging, they do not address the possibility of proteasome poisoning by accumulating levels of insoluble proteins. Therefore, it seems prudent to perform some studies to investigate proteasome assembly and catalytic activity, as well as polyubiquitination of Htt-polyQ in the presence and absence of eff. The proteomic studies are essentially phenomenological, and do not provide mechanistic insight because none of the hits was experimentally characterized. The authors do not explain why they selected eff for further characterization, rather than Ubc6 which appeared to have a stronger effect on Htt GFP accumulation (Fig. 1).

In Fig 3E the authors should indicate on the figure what regions of the gel/lane was quantified (3F). Same for Fig. 4.

In the lifespan studies (Fig 4D, E), why is the median lifespan 50 days versus 70 days for the controls?

Rev. 2:

In this manuscript, Hunt et al. performed a comprehensive genetic study on ubiquitin-conjugating enzymes (E2s) using the Drosophila retina Htt-polyQ model. Based on the results of this targeted RNAi screen, they decided to focus on UBE2D/eff, which was among the few E2s that, upon knock-down, increased Htt-polyQ aggregation. They then tested the role of UBE2D/eff in skeletal muscle and provided compelling genetic, biochemical, and proteomics data indicating the age-associated decline of UBE2D/eff protein levels contributes to the loss of proteostasis in aging and lifespan shortening. Transgenic expression of human UBE2D2 partially rescued the lifespan deficit and proteomic remodeling resulting from eff RNAi, supporting an evolutionarily conserved role of UBE2D in maintaining proteostasis during aging. Overall, this study was carefully designed and performed, and the manuscript was well-written. Nevertheless, the authors should consider addressing the following comments to strengthen the paper further.

Major points:

1. The authors performed careful RNAi analysis by including multiple RNAi lines targeting a single corresponding gene (Fig. 1B&C). They conclude that the variability is likely due to the efficacy of different RNAis. This should be at least tested for eff by RT-qPCR (as in Fig. 4C), which will justify the authors' choices of RNAi in the following studies. In Fig.1B, among the 7-8 RNAis against eff, only 2-3 significantly increased Htt72Q-GFP. Do those lines have the best RNAi efficiency compared to the others? Are they used in the following studies in muscle?

2. The p62 levels increase upon eff RNAi in the detergent-soluble fraction (Fig.3E&F, Fig.4 A&B), and likely in the detergent-insoluble fraction as well based on Fig. 4A&B. Do the data suggest an impairment of autophagy? The author concluded that the result supports the role of eff in protein quality control without providing a detailed explanation.

3. A difference between the eff-RNAi-induced proteins in Fig. 5F and 5G is: proteins in Fig. 5F decreased upon transgenic expression of UBE2D2 (without eff RNAi), while proteins in Fig.5G did not change. This supports the authors' idea that proteins in 5G are a protective response. The author could also test if the induction of proteins in 5G occurred at the mRNA levels (esp. the chaperones as part of the heat shock response).

Minor points:

1. The authors used p62/Ref(2)P and Ref(2)P/p62 at different places. Similarly, they used both UBE2D/eff and eff/UBE2D. It is better to keep the nomenclature consistent throughout the manuscript.

2. The labeling of sample names in Fig.4A (eg. "control + mCherry") is very busy.

3. In Fig. 5F, although those eff target genes likely increase protein levels upon eff RNAi due to "improper eff/UBE2D-mediated turnover", the authors did not formally rule out the possibility that the changes could occur at protein synthesis. They may want to soften the statement.

4. In discussion, the authors proposed that upon eff RNAi, misfolded proteins may clog or burden the proteasome, which is a very reasonable idea. Given the role of eff in mitophagy, eff RNAi may also interfere with the autophagy-lysosome system contributing to the loss of proteostasis.

---

## [Decision Letter · Decision Letter 2]

20 Dec 2024

Dear Dr Demontis,

Thank you for the submission of your revised Research Article entitled "The ubiquitin-conjugating enzyme UBE2D maintains a youthful proteome and ensures protein quality control during aging by sustaining proteasome activity" for publication in PLOS Biology. Your manuscript has been assessed by the PLOS Biology editors, the Academic Editor and by the two original reviewers, who are fully satisfied (see the reviews below).

On behalf of my colleagues and the Academic Editor, Ursula Jakob, I am delighted to let you know that we can in principle accept your manuscript for publication, provided you address any remaining formatting and reporting issues. These will be detailed in an email you should receive within 2-3 business days from our colleagues in the journal operations team; no action is required from you until then. Please note that we will not be able to formally accept your manuscript and schedule it for publication until you have completed any requested changes.

Please note that we have removed 'eff' from the title, as we think it is better to name there only the most common name of the protein.

PRESS

Sincerely, 

Ines

--

Ines Alvarez-Garcia, PhD

Senior Editor

PLOS Biology

Reviewers' comments

Rev. 1:

I have reviewed the manuscript from Hunt et al describing the role of the UBE2D, an E2 enzyme, in regulating protein quality control. The authors have satisfactorily addressed my concerns, and the revised paper is convincing and well written. This is an important piece of work, which offers a new view of proteasome regulation by an E2 enzyme. New studies showing that autophagy is unaffected by blocking eff expression supports a more specific role in the regulating proteasome activity.

This study meets the high quality expected for publication in PLOS Biology and I enthusiastically support its publication.

Rev. 2:

The authors included new data and edits, which adequately addressed my comments and strengthened the manuscript. The current manuscript is suitable for publication at PLOS Biology.